# EEG and ERP biosignatures of mild cognitive impairment for longitudinal monitoring of early cognitive decline in Alzheimer's disease

**Amir H. Meghdadi**[1]*, **David Salat**[2,3], **Joanne Hamilton**[4], **Yue Hong**[2], **Bradley F. Boeve**[5], **Erik K. St Louis**[5,6], **Ajay Verma**[7], **Chris Berka**[1]

1 Advanced Brain Monitoring, Inc., Carlsbad, CA, United States of America, 2 Massachusetts General Hospital, Boston, MA, United States of America, 3 Harvard Medical School, Boston, MA, United States of America, 4 Scripps Health, La Jolla, CA, United States of America, 5 Departments of Neurology and Medicine, Division of Pulmonary and Critical Care Medicine, Center for Sleep Medicine, Mayo Clinic College of Medicine and Science, Rochester, MN, United States of America, 6 Department of Clinical and Translational Research, Mayo Clinic Health System Southwest Wisconsin, La Crosse, WI, United States of America, 7 Formation Venture Engineering, Boston, MA, United States of America

* amir@b-alert.com

**Data Availability Statement:** Below, please find the DOI for the repository where the data is uploaded: https://doi.org/10.7910/DVN/GDGIVG.

## Abstract

Cognitive decline in Alzheimer's disease is associated with electroencephalographic (EEG) biosignatures even at early stages of mild cognitive impairment (MCI). The aim of this work is to provide a unified measure of cognitive decline by aggregating biosignatures from multiple EEG modalities and to evaluate repeatability of the composite measure at an individual level. These modalities included resting state EEG (eyes-closed) and two event-related potential (ERP) tasks on visual memory and attention. We compared individuals with MCI (n = 38) to age-matched healthy controls HC (n = 44). In resting state EEG, the MCI group exhibited higher power in Theta (3-7Hz) and lower power in Beta (13-20Hz) frequency bands. In both ERP tasks, the MCI group exhibited reduced ERP late positive potential (LPP), delayed ERP early component latency, slower reaction time, and decreased response accuracy. Cluster-based permutation analysis revealed significant clusters of difference between the MCI and HC groups in the frequency-channel and time-channel spaces. Cluster-based measures and performance measures (12 biosignatures in total) were selected as predictors of MCI. We trained a support vector machine (SVM) classifier achieving AUC = 0.89, accuracy = 77% in cross-validation using all data. Split-data validation resulted in (AUC = 0.87, accuracy = 76%) and (AUC = 0.75, accuracy = 70%) on testing data at baseline and follow-up visits, respectively. Classification scores at baseline and follow-up visits were correlated (r = 0.72, p<0.001, ICC = 0.84), supporting test-retest reliability of EEG biosignature. These results support the utility of EEG/ERP for prognostic testing, repeated assessments, and tracking potential treatment outcomes in the limited duration of clinical trials.

**Funding:** [This work is supported by National Institute of Health Grant numbers R44AG050326, R44AG054256 as well as NIA ADRC grant numbers P30 AG06267] The funders had no role in study design, data collection and analysis, decision to publish, or preparation of the manuscript.

**Competing interests:** AM and CB are employees of Advanced Brain Monitoring. Chris Berka is co-founder and shareholder of Advanced Brain Monitoring. Advanced Brain Monitoring is a commercial medical device manufacturer specializing in the acquisition and analysis of EEG during wake and sleep. This does not alter our adherence to PLOS ONE policies on sharing data and materials.

## Introduction

The prevalence of Alzheimer's disease (AD) and related dementia is increasing [1], with no existing curative treatment, and significant burden on both patients and caregivers [2]. Decades after the original amyloid cascade hypothesis [3], our understanding of the AD etiology is still evolving [4–7]. However, it is widely accepted that AD is defined on a progressive continuum. It starts with complex pathological changes in the brain with accumulating and cascading effects. These gradual pathological changes can lead to prolonged preclinical and prodromal stages before finally reaching the onset of significant clinical manifestations [8]. In this model, deposition of amyloid beta is considered the crucial factor in starting this chain of events. As such, characterization of AD and its severity has been mainly focused on the two ends of this causal chain. On one end, modern neuroimaging techniques and fluid biomarkers are now the gold standard in diagnosis of AD by detecting the underlying pathology. On the other end, clinical manifestations of AD and its severity is still assessed by measures of cognitive performance in neuropsychological testing/clinical rating scales [9–12]. However, this causal link between the original pathophysiology and the resulting cognitive symptoms is indirect, complex and takes a long time to emerge.

Recent trials of anti-amyloid treatments for early AD [13,14] have been successful in eliminating amyloid beta from the brain with only moderate effects on slowing cognitive decline [14] suggesting a lack of direct correlation between cognition and amyloid burden. Moreover, the expense of beta-amyloid imaging and invasiveness of cerebrospinal biofluid measurement are potential limitations toward widespread application. Therefore, there is an unmet need for biomarkers that are closely linked to cognition, non-invasive, and affordable compared to molecular or imaging biomarkers.

Synaptic dysfunction/loss [15–20] is likely the key link between molecular pathology and concordant clinical manifestations. Neuroimaging modalities such as fluorodeoxyglucose positron emission tomography (FDG-PET) or single-photon emission computed tomography (SPECT) are used for detecting focal and global neural hypometabolism and dysfunction. However, these methods are expensive, lack temporal resolution, and are only indirectly associated with the neural correlates of cognition due to the time lag between real-time neural activity and neural metabolism. Overall, there is a need for scalable progression biomarkers in clinical trials and therapeutic monitoring, and for screening assessment within large community dwelling populations.

An inexpensive alternative approach to directly measure biosignatures of cognitive decline in real-time is electroencephalography (EEG) and event-related potentials (ERP). EEG and ERP have been previously shown to be useful biosignatures of cognitive decline in Alzheimer's disease [20–31]. EEG waveforms represent aggregated post synaptic potentials measured at the scalp, providing a direct functional measure of neural activity in real-time with optimal temporal resolution. When applied during cognitive tasks in an ERP paradigm, EEG-based ERP measures represent the averaged, summated neural activity underlying specific cognitive processes such as memory and attention.

Fig 1 demonstrates a simplified diagram summarizing the chain of AD causation and different biomarker modalities. In 2018 National Institute of Health (NIH) and Alzheimer's Association proposed a research framework for biological definition of Alzheimer's disease [32] as an ATN(C) classification where AD biomarkers can measure A (amyloid beta deposition), T (pathologic Tau), N (neurodegeneration), and C (clinical manifestations). In this framework, neurodegeneration biomarkers detect late-stage effects and may not be suitable for measuring synaptic dysfunction at earlier stages. There is a need for biomarkers linked to early-stage neural dysfunction.

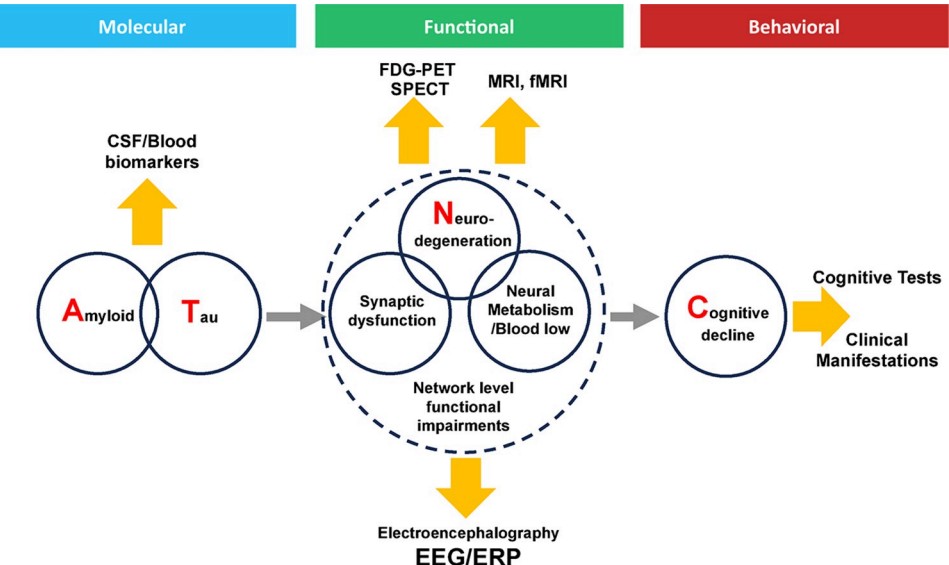

**Fig 1. A hypothetical simplified diagram of the chain of causation in Alzheimer's disease etiology and biomarkers as per the ATN(C) framework.** Wide arrows represent biomarker measurement modalities, while narrow arrows represent putative causation. Amyloid and tau pathologies (detectable by CSF molecular biomarkers) result in neural dysfunction/degeneration, which cause cognitive decline and other clinical manifestations. The triad of synaptic dysfunction, metabolic change and neurodegeneration are the basis of overall changes in neural functioning. Biomarkers based on FDG-PET, SPECT, and fMRI (i.e., topographical biomarkers) can detect localized neural dysfunction indirectly via hypometabolism or hyperperfusion. However, cognition emerges from interacting broadly distributed neural networks in real time. The modality of EEG/ERP is unsurpassed for assessment of functional neural impairment in real time.

Another theoretical model (A-T-P-N-O) has proposed that EEG/ERP assesses an added modality P (pathophysiology), and clinical manifestations (such as cognition and wake-sleep abnormalities) are referred to as O (output). EEG has emerged as a key biosignature for neuro-degenerative disease assessment (see [33,34]).

EEG and ERP signatures of Alzheimer's disease at the group [20–31] and individual level have been richly studied [35–37]. However, the utility of these measures for tracking disease progression (longitudinal study) at early stages (MCI) has received comparatively little previous attention.

Our aims were: (a) to determine if standalone resting state and ERP measures of MCI can be combined into a multi-modal unified measure, and to assess sensitivity and repeatability of this biosignature at an individual level; and (b) to analyze suitability of this biosignature for tracking trajectory of cognitive decline as a potential biomarker.

## Materials and methods

### Overall study approach/design

We compared a cohort of individuals diagnosed with MCI to an age-matched control group with respect to several EEG/ERP signatures. We used normalized effect size of the difference between the group means where higher effect sizes indicate better separation between MCI and controls at the group level. We used machine learning to build a classification model that combines these multiple EEG/ERP measures to provide a single unified score of MCI. We validated the model by ensuring a separate training and testing set. To assess the repeatability of

**Table 1. Baseline visit demographics: Participants demographics and cognitive scores at baseline visit.**

| | Baseline Visit | | | | |
|---|---|---|---|---|---|
| | Group | n | Age | %Female | MMSE |
| Total dataset[1] (n = 125) | HC | 57 | 68±5 (60–84) | 49 | 29.6±0.98 (25–30) |
| | MCI | 68 | 71±8 (53–88) | 37 | 27.1±2.81 (21–30) |
| Baseline dataset[2] (n = 82) | HC | 44 | 67±5 (60–84) | 52 | 29.7±0.57 (28–30) |
| | MCI | 38 | 69±7 (53–82) | 37 | 27.9±2.73 (21–30) [3] |

[1] Total dataset: Participants that completed at least resting state portion of the protocol at their baseline visit. [2] Baseline dataset: Participants that have available data for all the tasks at their baseline visit. Baseline dataset is later partitioned into Training dataset and Testing Data set (Table 2). [3] Three participants had missing MMSE data.

the proposed biosignatures, we compared the testing data at both baseline and a 1-year follow-up visits and calculated test-retest reliability of the composite score.

## Participants

Volunteer participants between the ages of 60–90 years were recruited across four sites in the United States. Advanced Brain Monitoring (ABM) in Carlsbad, California, USA (n = 44); Advanced Neurobehavioral Health (ANH) in San Diego, California, USA (n = 29); Massachusetts General Hospital (MGH) in Boston, Massachusetts, USA (n = 45); and Mayo Clinic (MAYO) in Rochester, Minnesota, USA (n = 8). All participants were selected after completion of a screening phone interview and a comprehensive in-office clinical and personal information questionnaire. Institutional Review Boards (IRB) at each site approved the study protocol (Advara IRB and Mass General Brigham Institutional Review Boards). All participants provided written informed consent. EEG acquisitions were scheduled in the mornings and participants were instructed to refrain from caffeine before the experiment. All data were collected between Feb 28, 2017, and May 25, 2021.

Table 1 shows selected demographics for both total data set (participants for whom at least resting state data is available) and baseline data set (participants for whom complete data for all tasks is available at the baseline visit).

A subset of participants also completed a follow-up visit approximately 1-year after baseline. The demographic table for participants who had complete data at the follow-up visit is shown in Table 2.

**Table 2. Training and Testing data sets: Participants demographics and cognitive scores for the two disjoint sets of Training and Testing data.** Testing data consists of participants with complete data available at both their baseline and follow-up visits.

| dataset | visit | Group | n | Age | %Female | MMSE |
|---|---|---|---|---|---|---|
| Training dataset (n = 35) | Baseline | HC | 15 | 68 ± 6 (60–84) | 60 | 29.7 ± 0.70 (28–30) |
| | | MCI | 20 | 69 ± 7 | 35 | 27.3 ± 3.04 (21–30) [2] |
| Testing dataset (n = 47) | Baseline | HC | 29 | 67±5 (60–82) | 48 | 29.8 ± 0.51 (28–30) |
| | | MCI | 18 | 69±7 (57–81) | 39 | 28.5 ± 2.35 (21–30) |
| | 1-Year Follow-up [1] | HC | 29 | 68 ± 5 (61–83) | 48 | 29.8 ± 0.38 (29–30) |
| | | MCI | 18 | 70±7 (58–82) | 39 | 26.1±3.12 (21–30) |

[1] Grouping based on the baseline diagnosis not reflecting changes from MCI (at baseline) to HC (at follow-up) for two of the MCI participants. [2] Three participants in the training data had missing MMSE scores. [3] Numbers reported as: mean ± SEM (mean-max) where SEM is standard error, and (min-max) is the range of values.

**Individuals with MCI.**  MCI participants at ANH: Volunteer participants with MCI at Advanced Neurobehavioral Health (ANH) study site were recruited from the greater San Diego area (USA) from either 1) a pool of individuals who enrolled in an Alzheimer's Disease Research Center (ADRC) longitudinal study or 2) a pool of individuals who were treated by community neurologists. Diagnostic criteria for MCI (amnestic type) were based on the neuropsychological evaluation adhered to DSM-5 criteria for minor neurocognitive disorder as follows: a) presence of objective cognitive impairment ($\geq$1.5 standard deviations) in the memory domain b) absence of decline in activities of daily living due to cognitive impairment, and c) absence of other medical or mental disease that explained the syndrome.

Eligible participants from the ADRC were diagnosed by two senior staff neurologists based on criteria developed by the National Institute of Neurological and Communicative Disorders and Stroke and the Alzheimer's Disease and Related Disorders Association (NINCD-S-ADRDA McKhann et al, 1984, and revised in 2011). The ADRC diagnostic procedure has been extensively documented (see [38]).

MCI participants at MGH: Volunteer participants at MGH were drawn from a longitudinal cohort study of brain aging and cognition. Participants were generally neurologically and psychiatrically healthy as determined by a medical screen and a neurological evaluation and exhibited broadly normal global cognitive functioning at the time of the assessment (MMSE: 24–30).

All the MGH participants were given a comprehensive neuropsychological evaluation battery containing 14 neuropsychological tests assessing global cognitive functioning, premorbid intelligence, and four specific cognitive domains. All tests were scored using standardized norms adjusted for demographic variables including age, sex, and education level. Sixteen standardized performance scores from the ten tests assessing the four specific cognitive domains were used to make neuropsychological classification. Diagnosis of MCI was determined following the Peterson criteria [39,40] that includes the presence of objective cognitive impairment ($\geq$1.5 standard deviations) in at least one cognitive domain. This criteria is slightly different than our prior work [23] for better standardization across the multiple sites, higher sensitivity of detecting MCI in terms of number of domains of impairment, and higher specificity in detecting MCI in terms of the level of impairment (1.5 standard deviation below normal). At the baseline visit, n = 31 participants were diagnosed as MCI at this study site.

MCI participants at MAYO: Volunteer participants at MAYO were drawn from the Mayo Clinic Alzheimer's Disease Research Center (Mayo ADRC), which is a longitudinal cohort study enrolling those with MCI or dementia. Participants at MAYO underwent neurologic and neuropsychologic evaluations, and diagnostic assessments, in a manner essentially identical to those stated above at MGH. Eight participants were diagnosed as MCI and enrolled in this protocol at this study site.

**Exclusion criteria.**  Participants were excluded if they reported any of the following conditions: known neurological or psychiatric disorders, cardiac arrhythmias, heart failure (e.g. myocardial infarction), epilepsy, HIV+ diagnosis, bipolar disorder, or major depression. Participants were not excluded for controlled hypertension, diabetes, high cholesterol, treated mild to moderate sleep apnea, or mild depression. Medical marijuana use was not a cause for excluding a participant, however, current usage amounts and frequency of use were documented.

## Experimental protocol

Participants completed a clinical and personal information questionnaire that included medical history and a medication inventory.

EEG and ECG (electrocardiogram) data were acquired using STAT™ X24 (Advanced Brain Monitoring, Carlsbad, CA), wireless EEG system (20 channels of EEG, 10–20 montage, linked mastoids reference, sampling rate 256 Hz, amplifier bandwidth 0.1–100 Hz). One channel of ECG was recorded with sensors placed on the right and left clavicles. STAT™ X24 uses passive Ag/AgCl electrodes with flexible circuit cables printed on a polyester strip. Each channel location contact is supported by a sponge donut filled with a conductive cream (Kustomer Kinetics, Arcadia, CA) with no need for skin abrasion. Technicians were instructed to keep the electrode impedances at or below 40kOhm [41] as per the manufacturer's guideline. The participants were seated in a comfortable chair with a laptop or desktop computer positioned on the desk directly in front of them. The computer screen was situated approximately 65cm away from the participant's face. EEG data were acquired continuously while the participants completed all the computer-based tasks using a structured acquisition software platform that provided instruction and administered each task in the pre-defined order. After completion of neuropsychological tests and answering personal and medical history questionnaires, the study protocol was conducted as follows: 5-minutes of resting state with eyes open, 5-min of resting state with eyes closed, a 20-min 3-choice vigilance task (3CVT), a 10-min standard image recognition (SIR) memory task, and a verbal memory scan (VMS) task. Data from the VMS task are not included in this manuscript because of the low compliance rate in our cognitively impaired group. During the eyes open task, participants were instructed to stare directly at a black fixation cross located in the center of a gray background. During the eyes closed task, they were instructed to close their eyes while maintaining wakefulness. ERP tasks included instructions followed by a short practice test before each task. Participants who failed the practice test 3 times were directed to the next task. The order of the tasks was the same for all participants.

## Neurocognitive tasks

**3CVT:** 3-choice vigilance task (3CVT) is a continuous performance task that probes sustained and selective attention [42,43]. 3CVT has been previously used in various applications for assessment of alertness and cognitive performance, mild cognitive impairment [44], HIV-associated neurocognitive disorder (HAND) [45], and cognitive performance decline due to sleep deprivation [46].

3CVT is an event-related potential (ERP) task that consists of a sequence of 376 trials. (See Fig 2A). At each trial, a visual stimulus (a geometric shape) is displayed at a random location on a blue screen. The subject's task is to discriminate between two types of frequent and infrequent visual stimuli by pressing the left and right arrow key on the keyboard, respectively. There are 264 (70%) frequent stimuli (targets) depicting an upward triangle (▲), and 112 (30%) infrequent stimuli depicting either a downward triangle (▼ nontargets: 55 trials, 15%) or a diamond shape ( distractors: 57 trials, 15%). The trials are presented in a pre-defined sequence with no obvious pattern. Each stimulus is presented for 400 ms with increasing length of stimulus onset asynchrony (SOA) across 5-minute quartiles. SOA is onset to onset interstimulus interval. During the first quartile, the SOA ranges from 1.5 to 3 seconds interstimulus interval, increasing up to 6 seconds during the second and 10 seconds during the third and fourth quartiles. The subject's response and the reaction time is recorded for further analysis.

**SIR**: Standard image recognition (SIR) task is a computerized working memory test (Fig 2B). SIR is a yes/no recognition memory test where the subject is first presented with a series of images (n = 20) to be memorized (*target* images). Subsequently, the subject is presented with a set of test images (n = 100) consisting of a random mix of the 20 memorized images (*target* stimuli) interspersed with 80 *novel* images. All *target* and *test* images belong to

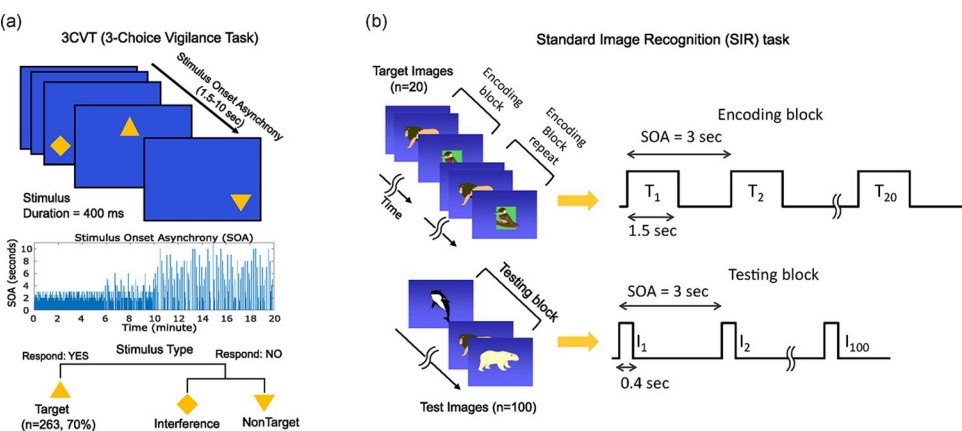

**Fig 2. ERP Neurocognitive tasks.** (a) 3CVT consists of a random sequence of 376 visual stimuli (trials). Each stimulus includes a yellow-colored geometric shape in a random location on an empty blue background. 70% of the trials are *target* trials containing an upward triangle, while the rest of the trials are either a downward triangle or a diamond shape; 15% each). The time between the onset of consecutive stimuli (SOA) varies ranging from 1.5 seconds to 10 seconds with values that are increasing during the 20-minute duration of the task. (b) SIR task consists of an *encoding block* of 20 *target* images that are presented every 3 seconds and remain on display for 1.5 seconds to be memorized. The *encoding block* is repeated twice before the start of a *testing block*, in which 100 stimuli (test images) are displayed once every 3 seconds. *Test images* contain the 20 memorized *target* images interspersed with 80 *novel* stimuli. Each testing stimuli remains on display for 400 milliseconds.

the same category e.g., pictures of either animals, or objects, or food, etc.) The subject's task is to respond by pressing the right or left arrow key on the keyboard to respond "yes" or "no" after each *test* image; responding "yes" to *target* (memorized) images and "no" to *novel* (previously unseen) images. SIR task starts with an encoding phase for memorizing the images. This phase consists of two identical back-to-back *encoding* blocks of trials (1 min each, for a total of two minutes). Each *encoding* block consists of the 20 *target* stimuli (images) presented in a sequence every 3 seconds and remain on the screen for 1.5 seconds (resulting in a fixed onset to onset inter stimulus interval ISI = 3 seconds for the encoding blocks). The subject is instructed to respond "yes" after seeing each stimulus. After the encoding phase is finished, a message is presented, and the subject is requested to press a key to proceed to the testing block of the task. The *testing* block consists of n = 100 trials, 20 of which are *target* images interspersed with n = 80 *novel* (previously unseen) images from the same category. During the testing block the ISI was kept at the same interval (3seconds) but the images were displayed on the screen for 400ms. The participant's task in the *testing* block is to respond "*yes*" to *target* images and "*no*" to novel images by pressing right/left arrow keys on the keyboard, respectively. The reaction time of the response and the percentage of correctly responded trials is recorded for further analysis.

## EEG preprocessing

For each experimental visit, EEG data corresponding to each task were segmented out and fed into a fully automated pre-processing pipeline that included (a) filtering in frequency domain, (b) automatic artifact rejection (exclusion), and (c) automatic artifact decontamination using independent component analysis (ICA). Filtering consisted of a bandpass filter (0.3-49Hz) for resting state data and (0.3–40 Hz) for ERP data, followed by a notch filter at 60 Hz to remove line noise. Artifact rejection (pruning) included automatic detection of invalid data segments (non-physiological and/or extreme artifacts) followed by either channel interpolation (if only

one channel was affected), or exclusion of the affected segments (artifact rejection). Invalid data segments were detected using a 2-seconds-wide moving window (50% overlap) over the time course of EEG signals. Each data segment (2-seconds wide) was marked for exclusion if either of the following criteria were met: (i) power of the signal in the data segment is 8 times larger than the average power across all channels; (ii) The amplitude of the signal remains more than 400uV for more than half of the segment or changing more than 500uV within the segment. These thresholds were chosen conservatively, in order to exclude only rare and extreme sections of data before applying artifact decontamination. For all ERP tasks, EEG data were epoched time-synced to stimulus onset (-0.8 to +1.3 seconds).

We used independent component analysis on continuous and epoched data in resting state and ERP tasks, respectively. We used EEGLAB software (v2023.0) [47] with default parameters for ICA decomposition (extended-ICA version of the logistic infomax ICA algorithm [48,49]) without dimension reduction. We used ICLabel [50] (v1.4) algorithm for assigning a classification probability to each component. ICLabel has been pre-trained using crowdsourcing and for each component provides the probability of being one of the following 7 categories: brain, muscle, eye, heart, line noise, channel noise, other (unknown). Subsequently, we rejected any component classified as a "brain" with less than 10% probability. This conservative threshold was chosen to ensure higher specificity as opposed to sensitivity in detection of non-brain components. Channel data were reconstructed after rejecting the non-brain components. We used the resulting clean data for PSD computation in the resting state task and for ERP analysis in the ERP tasks as described in the following two sections.

## PSD computations

We computed PSD at each channel similar to the method described in [23]. Briefly, the steps for computing the PSD are as follows. First, epoching data into overlapping (50%) one-second-long epochs (segments). Second, performing FFT (N = 256) on the windowed epochs (Kaiser window; b = 6) to compute the spectrum magnitude ($|X(f)|$) at each 1-Hz frequency bin ($f$) from 1–40 Hz. Unscaled power spectral density for each epoch is defined as: $psd = (1/Fs) \times 2|X(f)|^2$ where $Fs$ is the sampling frequency (256 Hz). Last, the $psd$ is averaged across the 3 overlapping epochs centered on each epoch and converting to logarithmic scale ($\log_{10}$) following by averaging across the 300 non-overlapping epochs of a resting state recording session. We defined PSD in each frequency band by averaging the above PSD values across all the frequency bins in a band. We defined frequency bands prior to this study and consistent with historical definitions across our previous studies as follows: delta (1–3 Hz), slow-Theta (3–5 Hz), Theta (3–7 Hz), slow-Alpha (8–10 Hz), Alpha (8–13 Hz), slow-Beta (13–20 Hz), Beta (13–30 Hz), Gamma (25–40 Hz).

## ERP analysis

Filtering, artifact rejection, epoching, and artifact decontamination were performed according to the procedure described above followed by ERP baseline removal using 100ms window before the stimulus onset. Epochs with EEG amplitude exceeding 100uV during the time window [–50 700] ms (relative to stimulus onset) were excluded to remove any potential remaining large artifact. Average ERPs for each stimulus type were computed by averaging EEG during correctly responded trials and after excluding artifact-contaminated trials. Any ERP with less than 9 trials (remained for averaging) was excluded. ERPs for each participant were analyzed (measured) and grand average ERPs were plotted. In the SIR task, we selected novel stimuli for further analysis and feature extraction because the number of clean and correctly responded trials (maximum 80 trials for 80 novel images) are higher than target trials

(maximum 20 trials for 20 memorized images) resulting in more robust ERP measures. As expected from our prior reports [44], SIR task is expected to elicit two distinct ERP components. An early component (P200) around 200 milliseconds after stimulus onset and a late positive potential (LPP) component consisting of a slow positive wave between ~400–800 ms post stimulus onset.

We computed ERP measures using both a classical hypothesis-driven method that measures specific ERP components as well as a data-driven method based on cluster-based permutation analysis. In the classical method of ERP measurements, we measured the amplitude and latency of the ERP components using an analysis window centered on the known ERP components previously identified for each task. ERP components were confirmed by visual examination of the grand averages and then components were measured at each EEG channel for each participant. We measured the average amplitude, the peak amplitude, and the latency of the peak amplitude within each analysis window.

As expected from our prior reports, 3CVT is expected to elicit an ERP morphology with an early component (P1) with a positive peak between ~100-200ms post stimulus onset and a late component (LPP: late positive potential) consisting of a slow positive wave between ~400–800 ms post stimulus onset. In the hypothesis driven analysis of ERP waveforms, we measured both latency of the early component (P1 in 3CVT and P200 in SIR) and average amplitude of the LPP component (for both tasks) at each EEG channel.

In the cluster-based method (described in the Statistical Analysis section below), we identified significant clusters in the time/channel space without the need for identifying components or window of analysis. In this method, we measured ERPs by averaging amplitude across the channel-time space corresponding to each significant cluster (cluster-based aggregated measure: CBA). In both ERP tasks we reported results for the frequent stimuli (*target* stimuli in the 3CVT task and *novel* stimuli in the SIR task).

## Statistical analysis

**Baseline comparisons.**   In order to identify significant differences between the MCI cohort and age matched controls at baseline, we performed both of the following: (a) hypothesis-driven parametric testing (t-test, uncontrolled, $p < 0.05$) to identify statistically significant differences in the mean of pre-selected frequency bands (for resting state EEG data) and pre-selected ERP measures (for ERP data) at each channel, hence resulting in multiple comparisons. (b) we also performed a data-driven cluster-based permutation analysis [51] to identify statistically significant clusters in the channel-frequency space (for resting state PSD data) and channel-time space (for ERP data).

Cluster-based permutation analysis: Cluster-based permutation analysis was chosen since it addresses the multiple-comparison problem, and yields a single aggregated measure derived from each significant cluster, providing a unified measure using multiple channels and frequencies/times. In the resting state PSD analysis, we defined the 2-D discrete space of channel-frequency as the 20 (EEG channels) x 40 (frequency Hz bins) space of 800 cells. In the ERP analysis we defined the discrete space of channel-time as the 20 (EEG channels) x 256 (discrete time points between -200 and +800 milliseconds after stimulus onset) space of 5000 cells. We then performed multiple *t-tests* between the two groups at each cell of the discrete space. We used $p < 0.01$ as an initial significance threshold and defined a cluster to be the set of all connected cells in the 2-D space that exceeded this threshold. We call the set of these clusters as *primary clusters*. We used *edge connectivity* type for identifying clusters *i.e.* two cells in the space belong to the same cluster if they belong to the same discretized value in either dimension (same channel or the same timepoint/frequency). We defined the *geometric size* of a

cluster as the number of cells in that cluster (hereby referred to as *size*) and the *statistical size* of the cluster (hereby referred to as *tsize*) as the sum of all *t* statistics for cells in the cluster. Minimum cluster size was set to 2 channel-Hz in resting state data and 20 (78 channel-milliseconds) in ERP analysis. Subsequently, we performed a permutation analysis where we randomly assigned participants to each group (shuffling the labels) 1000 times (iterations). In each iteration we identified the largest *random* cluster (using their statistical size: *tsize*) and obtained the distribution of the largest random cluster across all 1000 iterations. Finally, the statistical significance of each *primary* cluster was defined as the percentile value of that cluster *tsize* in the distribution of maximum *tsize*s across all random clusters. We used the 95[th] percentile as the threshold of statistical significance for the size of formed clusters, to exclude chance primary clusters. We visualized each cluster via (a) a 2-D map of the space that showed the extent of each cluster and its geometric size. (b) plotting the number of channels that belonged to the cluster at each value of the second dimension (frequency /or time), and (c) a head-map of all the channels that belonged to the cluster at any point and the largest frequency (/or time) window that covered the cluster. We used each cluster to derive a cluster based aggregated measure (hereby referred to as CBA) by averaging the PSD (in the case of resting state analysis) or the amplitude of the ERP (in the case of ERP data) across all the cells in a cluster resulting in a single measure to be used for further analysis.

**Feature selection.**   Since differences between the MCI and control groups occur across measurement modalities (resting state PSD, cognitive tasks ERPs/performance), EEG channels, and time points/frequencies, our feature selection process selected and aggregated group differences, providing a non-arbitrary, bias-free, data-driven, limited set of EEG/ERP features to differentiate MCI participants from healthy controls. We used 6 modalities of data including: (a) relative PSD in resting state, (b) absolute PSD in resting state, (c) 3CVT task ERPs, (d) 3CVT task performance, (e) SIR task ERPs, and (f) SIR task performance. In order to select features derived from (and representative of) all these modalities, we selected the features using the cluster based aggregated measure associated with the significant cluster(s) in each measurement modality. If none of the clusters reached the significance level (95[th] percentile threshold in the permutation analysis), then the cluster with the highest percentile value was selected (this happened only for absolute PSD modality). Additionally, and in order to include a feature that directly measures the latency shift (delay) of the ERP early components in MCI, we also added an additional feature based on the latency of a representative channel (Fz in the SIR task and Cz in the 3CVT task). Fz and Cz were chosen arbitrarily from the midline electrodes that show a prominent peak on grand average ERP waveforms. Last, for each ERP task we also included the two most important performance (behavioral) measures, average reaction time and accuracy (percentage of the trials with correct response). Consequently, the above 12 features (MCI predictors) were included in the final selection of a feature set (2 from resting state, 3 from 3CVT ERPs, 2 from 3CVT performance, 3 from SIR ERPs, 2 from SIR performance). We performed the feature selection for each modality separately and based on all available data for each modality at the baseline visits (total data set Table 1).

**Analysis of variance.**   We used *Baseline* data set (see Table 1) for analysis of variance (using all the participants who had complete data in all modalities at their baseline visits). For each one of the 12 features, we used ANOVA to analyze the main effect of group (MCI or HC), sex, or age (younger or older than 70 years old) and their interaction effect. 70 years old threshold was chosen in order to dichotomize the participants into sets of almost equal size.

<u>Analysis of consistency of individual predictors:</u> We assessed the correlation between individual features in the feature set by computing Pearson correlation coefficients, as well as intra-class correlation coefficient (ICC) measures of internal consistency using Cronbach's alpha equations [52].

**Classification.** We used the feature set described above for training a binary classification model that classified each participant as MCI or HC. We used a support vector machine classifier (SVM; linear kernel function with standardization of the features) as the classification model with positive class (classification score>0) indicating MCI. In order to minimize overfitting, we divided the *baseline* data set into a *training* and *testing* data set (see Table 2). Any participant with complete data in all modalities at both their baseline and their 1-year follow-up visits were assigned to the *testing* data set and the remaining participants (those with only baseline data available) were assigned to the *training* data set. This method of partitioning was performed to enable testing completely novel data (participants' data that have not been used in training the classifier) at both baseline and follow-up visit and hence assess the repeatability of the classification models for longitudinal assessment of individual patients. We used the area under the ROC curve (AUC) and the percentage of data classified correctly (accuracy) for assessing the performance of the binary classification. In each classification we used the normalized effect size of the difference between means of the classification scores in each group as a measure of separability between groups using a classification model. We assessed the performance of this trained classifier in all the following three ways with the $3^{rd}$ method properly reflecting the performance of the classification approach: (a) the re-substitution accuracy (suffers from overfitting) of the model when applied to the same training data (b) cross validation (cv) accuracy (5-fold cv) and the associated AUC, and (c) full validation of the model by testing completely novel data (accuracy and AUC) at both baseline and follow-up visit separately. An ideal classifier is assessed by both its accuracy and stability (repeatability) when tested on novel data at two time points. We computed the Pearson correlation coefficient comparing the MCI classification scores between baseline and the follow-up visit. Additionally, and in order to explore the performance of a classifier trained on a larger set, we also trained another SVM classifier using all the *baseline* dataset (*i.e.*, all baseline visit data from either *training* or *testing* data sets) and validated with a 5-fold cross validation. This classifier was not used for assessment of the repeatability of the classification scheme.

In all the above assessments, we used three methods for identifying the classification performance, the area under the ROC curve (AUC), accuracy (ACC: percentage of correctly classified data points), as well as balanced accuracy (BAC: arithmetic mean of the sensitivity (true positive rate) and specificity (true negative rate)) at the operating point.

We assessed the predictive power of each EEG/ERP predictor, using both SVM weights in the classifier as well as a *permutation-feature-importance* method. In the first method, we computed the relative importance of each feature (predictor) in the SVM classifier by dividing the absolute value of the corresponding weight to the sum of the absolute values of all weights (SVM with linear kernel function). In the permutation feature method, for each predictor, we randomly shuffled the values of the predictor across all data points and assessed the performance of the classifier on the shuffled data (repeating this 100 times and averaging the performance across all 100 iterations). We used the average reduction in performance (using either of performance measures) as a measure of the feature importance (the higher the predictive power, the higher performance drops if the predictor values are shuffled).

Finally, we explored the performance of different classification models (linear discriminant analysis, logistic regression, K nearest neighbors, and decision tree classification) by computing the accuracy of 5-fold cross validation on this baseline data. (See S1 Table).

<u>Statistical significance:</u> Unless otherwise explicitly described, the threshold for statistical significance was set to $p = 0.05$ (and to $p = 0.01$ for first-level statistics in the cluster-based permutation analysis). p values have been reported as follow: $p>0.05$ (the actual p-value was reported), $0.01<p\le = 0.05$ ($p<0.05$; also denoted by *), $0.001<p\le0.01$ ($p<0.01$; also denoted

by **), p≤0.001 (p<0.001; also denoted by ***). We used Hedges' *g* for computing normalized effect size.

Missing data and outliers: For each modality of data (resting state, 3CVT task, SIR task) all participants with available data were included in analysis. All data exclusions (artifact rejection, trial rejection, excluding ERPs with low number of trials) were performed using the fully automated analysis pipeline. Outliers in the data (e.g., extremely poor behavioral performance in ERP tasks, etc. were not excluded to reflect the variability in the real-world application of the methods). For multi-modal analysis (classification and providing composite score) only participants with complete data in all modalities were included for analysis. For longitudinal analyses, only participants with complete data at both baseline and follow-up visits were included.

## Results

### Data quality

In the HC group, 97% of the participants completed all tasks while this percentage in the MCI group was 60%. On average, the percentage of data that were usable after artifact rejection and interpolation were 97%, 96%, and 95%, in resting state, 3CVT task, and SIR task, respectively. The number of rejected components in ICA decontamination on average were 4.5, 5.0, and 4.8 components for each of the 3 tasks, respectively. After excluding incorrectly responded and artifact-contaminated trials, on average 82% and 56% of ERP trials were included for averaging in the 3CVT and SIR tasks, respectively. 1.8% and 3.0% of ERP acquisitions were excluded due to low numbers of ERPs (<9 trials) for 3CVT and SIR tasks, respectively. Only target stimuli in 3CVT and novel stimuli in SIR tasks were used to compute these statistics. These data quality statistics were obtained from a processing pipeline that preprocessed all data in the parent study that includes data from participants with other conditions or age groups not included in this manuscript.

### Baseline MCI characteristics

**Resting state eyes closed.** 55 HC (aged 60–84, mean (sd): 67.6(5.3) years) and 67 MCI (aged 53–88, mean(sd): 71.4(8.1)) participants completed resting state EEG with eyes closed at baseline. Overall relative and absolute PSD graphs (averaged across all channels) are shown in Fig 3A. Hypothesis driven analysis of the PSD included group differences between the HC and MCI group at each channel and each frequency bands (e.g. Theta 3–7 Hz, Alpha 8–13 Hz, slow-Beta 13–20 Hz, and Theta to Alpha ratio (TAR)). Fig 3B shows topographical maps of the group differences in PSD measures comparing the HC and MCI group for selected frequency bands and the statistically significant channels (t-test; p<0.05, uncorrected). The MCI group exhibited reduced power in slow-Beta, increased power in Theta, and increased TAR compared to healthy controls. Normalized effect sizes for the most significant channel were -0.66, +0.45 and +0.39 for relative slow-Beta, Theta, and TAR, respectively. Alpha power was similar between groups (no significant difference).

Cluster-based permutation analysis of relative PSD data resulted in a significant cluster (99[th] percentile) in the frequency-channel space (cluster 1: Fig 3C), while the most significant cluster in the absolute PSD data did not reach significance (89[th] percentile) (cluster 2: Fig 3E). The channel-frequency coverage of the two clusters was consistent with the Beta reduction and Theta increase. Cluster-based aggregated measures (CBA) of resting state PSD (PSD averaged across the channel-frequency cluster) for cluster 1 and 2 are shown in Fig 3D and 3F, respectively. The normalized effect size of the difference between the CBA in the MCI and HC groups was (ES = -0.70 and ES = 0.49) for relative and absolute PSD, respectively. The two

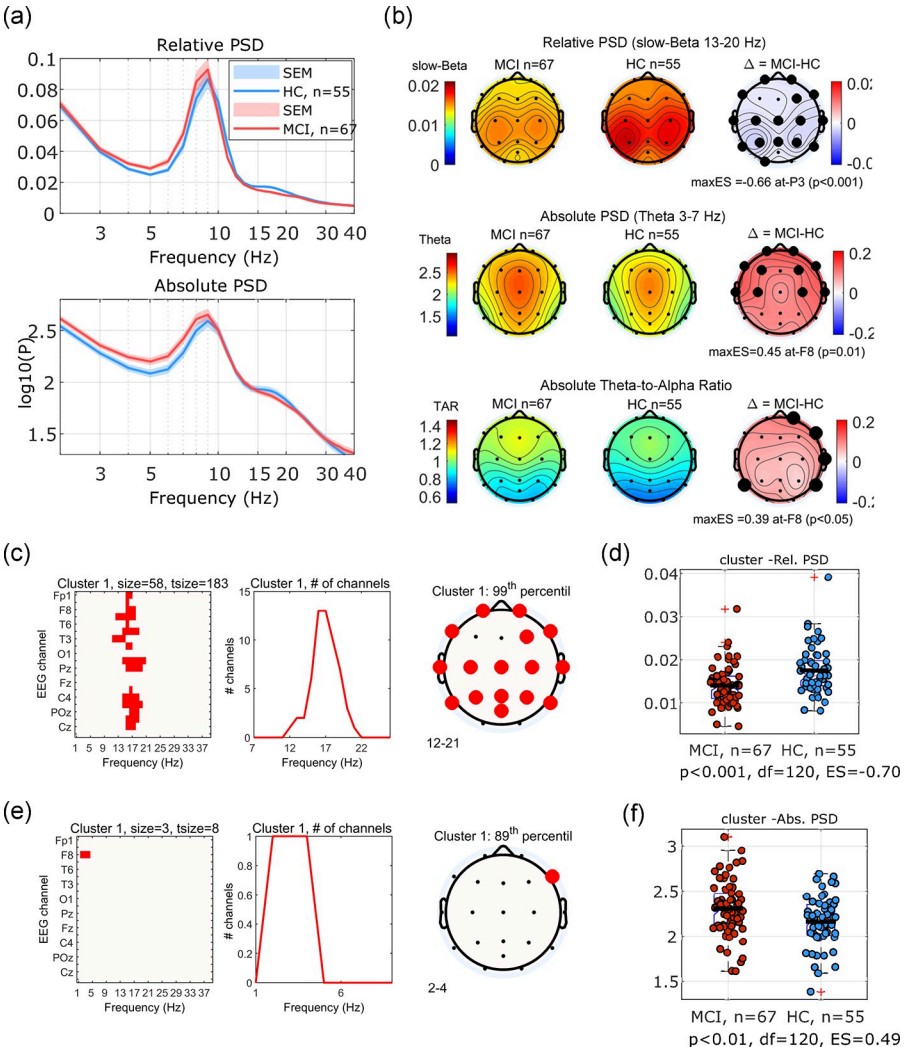

**Fig 3. Baseline visit resting state EEG with eyes closed.** (a) Group average relative (top) and absolute (bottom) PSD graphs (averaged across time and across channels). Shaded areas represent standard error of the mean. (b) Topographical maps of classical PSD measures from top to bottom: Relative PSD in slow-Beta band (13–20 Hz), absolute PSD in the Theta band (3–7 Hz) and Theta-to-Alpha ratio (TAR). Channels with statistically significant differences (t-test, p<0.05, uncorrected) are marked with black dots. (c) Significant cluster in the channel-frequency space for relative PSD after cluster-based permutation analysis capturing the effect in the Beta band (d) group differences between the HC and MCI group with respect to this cluster's CBA. (e) The largest cluster for the absolute PSD overlaps with the Theta band effect but it (89<sup>th</sup> percentile) did not reach the significance threshold. (f) group difference using the CBA measure derived from the most prominent (yet not significant) cluster for absolute PSD.

emerged clusters are consistent with the observed reduction of Beta and increased Theta power in the classical univariate analysis of frequency bands in each channel.

**Standard Image Recognition (SIR) memory task.** 46 HC (aged 60–84, mean(sd) 67.5 (5.4) years, MMSE: 28–30 mean(sd): 29.8(0.6), 52% female) and 43 MCI (aged 53–84, mean (sd): 70.1(7.4) years, MMSE: 21–30 mean(sd): 27.9(2.7), 35% female) participants completed the baseline SIR task. Grand average ERP waveforms for trials with *novel* stimuli at selected channels Cz and Pz are shown in Fig 4A. Same data for all channels are shown in the supporting information section (S1 Fig). Topographical maps of the measured components averaged across groups and the difference maps are shown in Fig 4B. Channels with significant

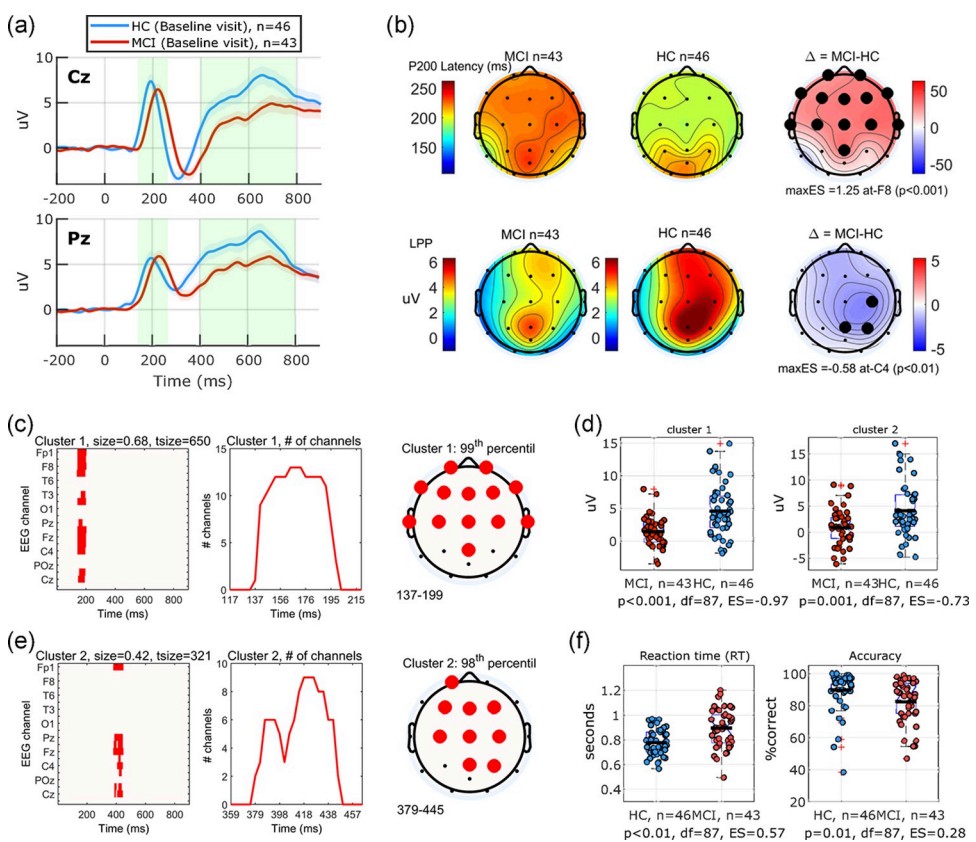

**Fig 4. Baseline visit (SIR task).** (a) Grand average ERP waveforms at baseline visit in response to novel stimuli in SIR task plotted for two EEG channels Cz, Pz as an example. Waveforms demonstrate the overall increased latency of the early peak (P200), and reduction of the amplitude of the late positive potential (LPP) component in the MCI group, compared to HC. (b) topographical maps of the classical measures of P200 and LPP components across the scalp, channels with statistically significant differences (t-test, p<0.05, uncorrected) between the two groups are circled with the largest effect size reported. (c),(e) Cluster-based permutation analysis revealed two significant (>95th percentile) clusters in the channel-time space. The first cluster extends over 0.68 channel-seconds covering the time window 137–199 milliseconds and the second cluster extends over 0.42 channel-seconds covering the time window 379–445 milliseconds. A 3rd cluster (not shown here) extending over 0.19 channel-seconds covering the time window 582–680 milliseconds and over channels C4,P4 and 94th percentile did not reach the 95th percentile significance threshold. (d) Amplitude of the ERP response averaged across each cluster mask shows significant group differences with normalized effect sizes ES = -0.97 and ES = -0.73 for the first and second clusters, respectively. On average, the MCI group showed significantly longer reaction time (+117 ms, ES = 0.57), and significantly lower accuracy of response (7%, ES = 0.28).

differences between the MCI and HC groups (t-test; p<0.05) demonstrated a delayed latency of the early component across all frontal and central channels and a decreased LPP amplitude across channels C4, P4, and PZ. Cluster-based permutation analysis revealed two significant clusters (99th and 98th percentile) associated with the early and late components (Fig 4C and 4E). The emerged clusters are consistent with the observed delayed latency of the early component and reduced amplitude of the LPP component in the classical method of measuring the components. Differences in cluster-based aggregated ERP measures between the two groups showed a normalized effect size ES = -0.97 and ES = -0.73 for the first and second cluster, respectively (Fig 4D). The MCI group on average exhibited slower reaction time (+117 milliseconds on average, t-test, p<0.01, ES = 0.57) and lower accuracy of responses (7% less on average, t-test, ES = 0.28). Fig 4F.

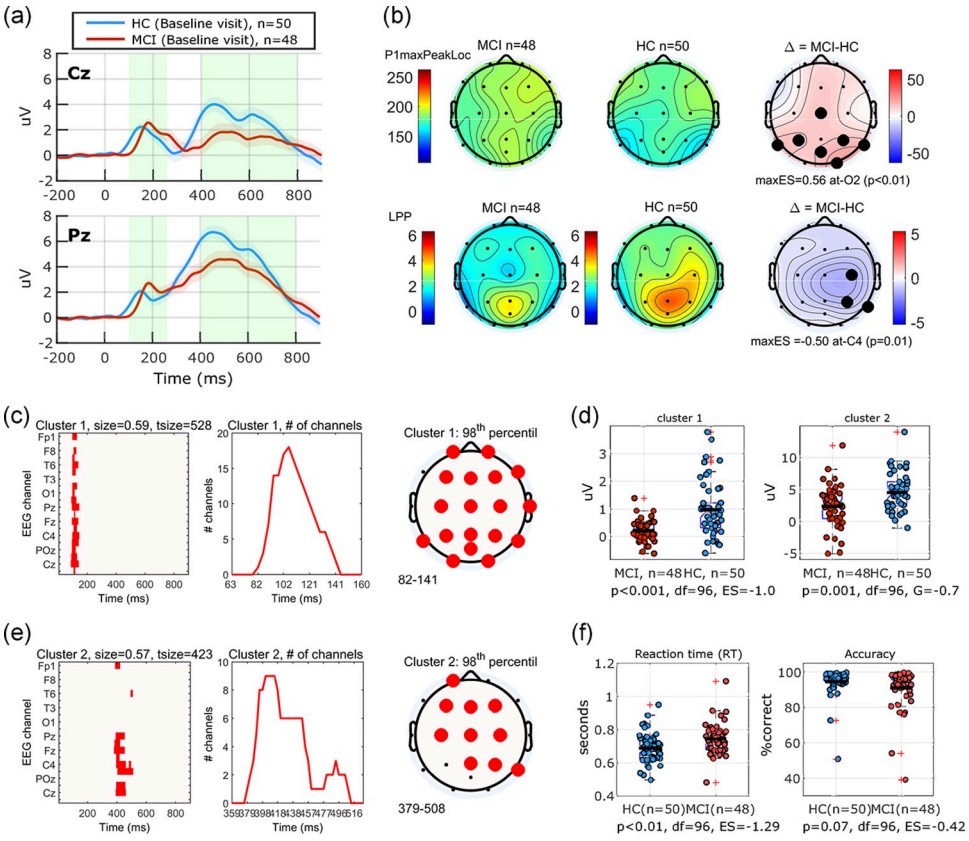

**Fig 5. Baseline visit (3CVT).** (a) Grand average ERP waveforms at baseline visit in response to frequent stimuli (Target) in 3CVT task plotted for two EEG channels Cz, Pz as an example. Waveforms demonstrate the overall increased latency of the early peak (P1), and reduction of the amplitude of the late positive potential (LPP) component in the MCI group, compared to HC. (b) topographical maps of the classical measures of P1 and LPP components across the scalp, channels with statistically significant differences (t-test, p<0.05, uncorrected) between the two groups are marked with black dots and the largest effect size reported. Cluster-based permutation analysis revealed two significant (>95th percentile) clusters in the channel-time space. (c) The first cluster extends over 0.59 channel-seconds covering the time window 82–141 milliseconds. (e) the second cluster extends over 0.57 channel-seconds covering the time window 379–508 milliseconds. (d) Amplitude of the ERP response averaged across each cluster mask shows significant group differences with normalized effect sizes ES = -1.0 and ES = -0.7 for the first and second clusters, respectively. (f) on average, the MCI group showed significantly longer reaction time (+55 ms, ES = 1.29), and a trend towards less accuracy (5%) that did not reach the threshold for statistical significance (p = 0.07).

**3CVT attention task.** 50 HC (aged 60–84, mean(sd) 67.9(5.6) years, MMSE: 25–30, mean (sd): 29.7(0.9)) and 48 MCI (aged 53–82, mean(sd) 69.3(6.9) years, MMSE: 21–30, mean(sd): 27.6(2.8)) participants completed the 3CVT task. Grand average ERP waveform for trials with *target* (frequent) stimuli at selected channels Cz, Pz are shown in Fig 5A. Grand averages for all channels are shown in the supporting information (S2 Fig). Topographical maps of the measured components averaged across groups and the difference maps are shown in Fig 5B. Channels with significant differences between the MCI and HC groups (t-test; p<0.05) demonstrate delayed latency of the early component across several channels in the posterior and central areas, and decreased LPP amplitude across channels C4, P4, and T6. Cluster-based permutation analysis revealed two significant clusters (98th and 98th percentile) shown in Fig 5C and 5E. These clusters are consistent with the early and late measurements. Differences in cluster-based aggregated ERP measures between the two groups are shown in Fig 5D. The normalized effect sizes of the difference between the CBA in the two groups were ES = -1.0 and ES =

**Table 3. Comparing individual EEG/ERP measures: Normalized effect size of the difference between the MCI and HC group and analysis of variance for each single EEG/ERP measure.**

| Variable Type | | EEG Resting PSD clusters | | ERP (3CVT task) | | | ERP (SIR task) | | | Performance 3CVT | | Performance SIR | |
|---|---|---|---|---|---|---|---|---|---|---|---|---|---|
| Variable | | Abs | Rel | Latency @Pz | cluster1 | cluster2 | Latency @ Pz | cluster1 | cluster2 | ACC | RT | ACC | RT |
| Normalized Effect Size (ES) | | +0.49 | -0.7 | +0.35 | -1.0 | -0.7 | +0.86 | -0.97 | -0.73 | -0.42 | +1.29 | -0.28 | +0.57 |
| p value (t-test) | | ** | *** | p = 0.08 | *** | *** | *** | *** | *** | p = 0.07 | ** | ** | ** |
| ANOVA Main effect $F_{1,75}$ | Dx | F = 8.1 ** | F = 5.2 * | F = 3.9 * | F = 17.5 *** | F = 8.7 ** | F = 24.9 *** | F = 15.8 *** | F = 7.7 ** | F = 2.1 - | F = 9.6 ** | F = 4.3 * | F = 28.9 ** |
| | Age | F = 1.2 | - | - | F = 1.3 | - | F = 3.9 | F = 1.2 | - | - | F = 1.6 - | F = 4.7 * | - |
| | Sex | F = 1.2 | - | - | - | F = 7.0 ** | F = 1.2 | - | F = 7.6 ** | - | - | - | - |

ACC: Accuracy, RT: Reaction Time, Abs (absolute PSD), Rel (Relative PSD)

* (p<0.05)

** (p<0.01)

*** (p<0.001).

Dx: Diagnosis group (MCI v. HC), The only significant interaction effect was between Sex and Diagnosis (F = 4.5, p<0.05) on RT in the SIR task.

-0.7 for the first and second cluster, respectively. The MCI group on average exhibited slower reaction time (55 milliseconds on average, t-test, p<0.01, ES = 1.29) when compared to the HC group. On average, MCI trended toward less accurate responses as measured by percentage of trials with correct responses (p = 0.07). Fig 5F shows the group differences in terms of these performance measures.

**Feature selection and analysis of variance.** Table 3 summarizes group differences between the MCI and HC group with respect to the individual EEG/ERP measures described in the previous section. Normalized effect sizes show the summary of results from the previous sections using all available data. ANOVA results show the F-statistics and significance of each main effect and interactions from an analysis of variance on the baseline data set (participants with complete data in all tasks).

ANOVA showed a main effect of diagnosis group (MCI vs. HC) for all 12 measures except response accuracy in the 3CVT task. Age had a significant effect on response accuracy in the SIR task (p<0.05, F = 4.7), and sex had a significant effect on the second cluster of both the SIR and 3CVT tasks. The only interaction was between sex and diagnosis group (F = 4.5, p<0.05) for the reaction time in the SIR task.

Correlations between each pair of the features are shown in Fig 6. Overall, cluster-based measures in both the 3CVT and SIR were highly correlated. Latency based measurements were also correlated between the two ERP tasks.

## Longitudinal changes and repeatability of EEG signatures

Fig 7 shows PSD at both the baseline and follow-up visits plotted for each group. Group average for relative PSD (Fig 7A) and absolute PSD (Fig 7C) at both visits are shown with solid (baseline visit) and dashed lines (1-year follow-up visit). Cluster-based permutation analysis of within-subject changes (based on a paired t-test for first level statistics) revealed no significant differences between the baseline and follow-up visits in either the HC or MCI group. CBA measures were correlated (ICC = 0.88, r = 0.79, p<0.001) for the relative PSD cluster (Fig 7B) and (ICC = 0.63, r = 0.47, p<0.001) for absolute PSD cluster (Fig 7D).

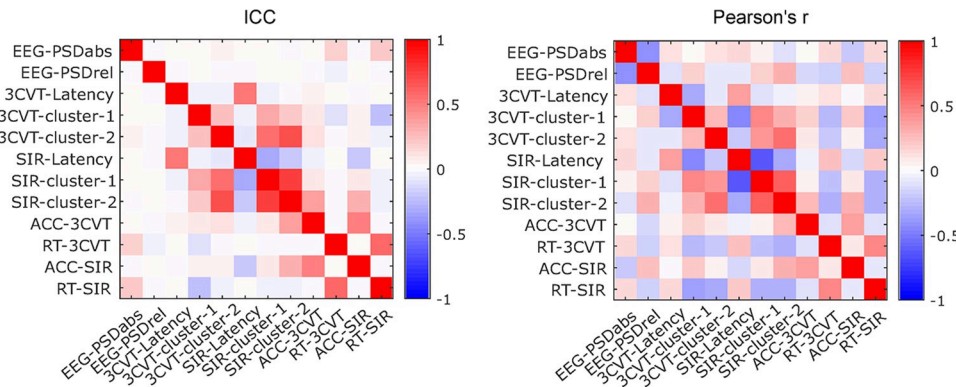

**Fig 6. Degree of consistency between predictors.** Degree of consistency between each pair of predictors using Cronbach's Alpha ICC (left) and Pearson's correlation (right).

Fig 8 shows the longitudinal results for the SIR task. Cluster-based permutation analysis revealed no significant differences between the baseline and follow-up visits in either the HC or MCI group. CBA measures did not significantly change from baseline to follow-up visit. Fig 8B shows the test-retest reliability of cluster-based measures (ICC = 0.85, r = 0.76, p<0.001) and (ICC = 0.85, r = 0.75, p<0.001) for the first and second cluster, respectively. The performance measures reliabilities were (ICC = 0.71, r = 0.55, p<0.001) and (ICC = 0.84, r = 0.74, p<0.001) for the reaction time and accuracy measure, respectively. Performance measures did not significantly change from baseline to follow-up visit.

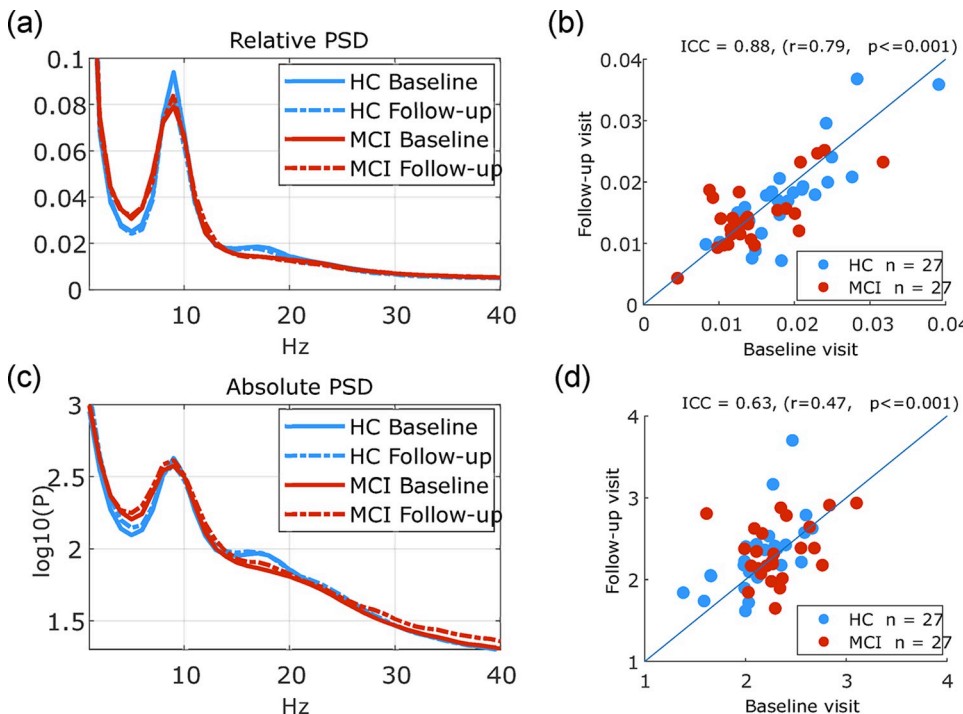

**Fig 7. Longitudinal changes in EEG resting state eyes closed PSD.** (a),(c) Group average relative and absolute PSD at baseline and follow-up visits for the HC and MCI participants that had resting state data at both visits. (b),(d) scatter plots showing the EEG measure based on relative PSD significant cluster (b) and absolute PSD cluster (d).

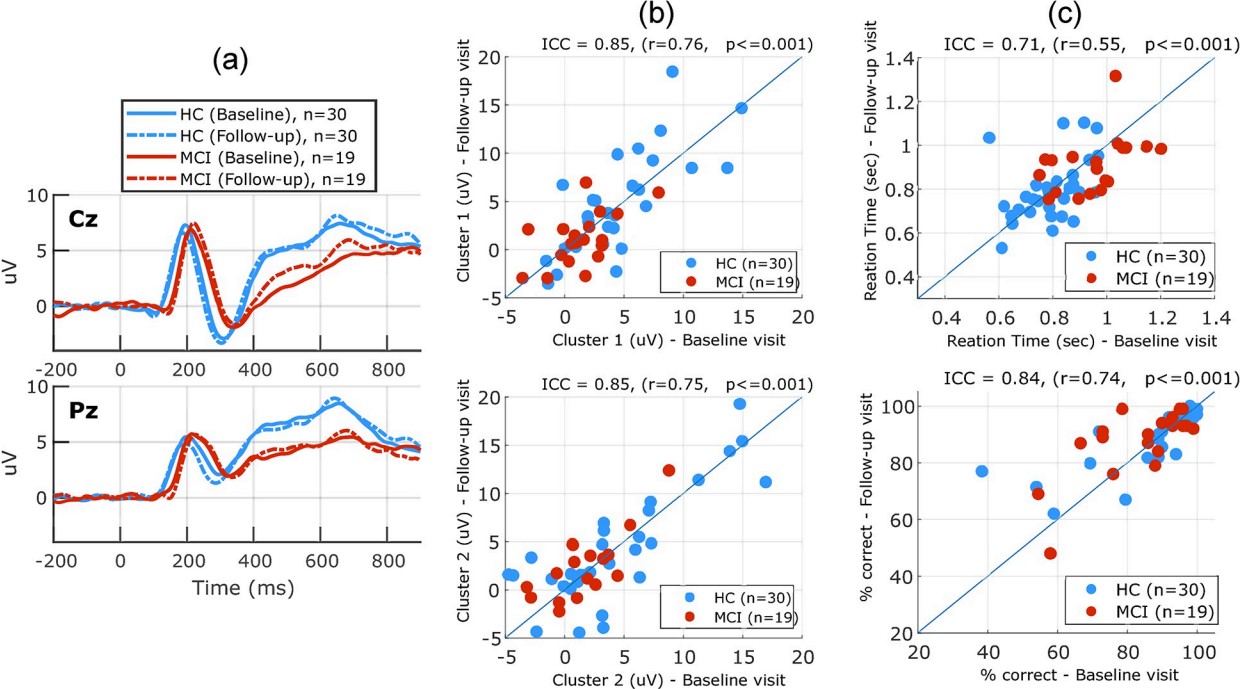

**Fig 8. Longitudinal changes in the SIR task.** (a) Grand average ERP waveforms (in response to novel stimuli in the SIR task) at both the baseline visit and 1-year follow-up, for the HC and MCI groups plotted for two EEG channels Cz, Pz as an example. (b) scatter plots showing the test-retest reliability of the EEG measures obtained using cluster 1 (top) and cluster 2 (bottom) in the SIR task. (c) scatter plots showing test-retest variability of the performance measures: reaction time (top) and percent correct (bottom) in the SIR task. Neither performance measures nor EEG measures significantly changed from baseline to follow-up visit in this task.

Fig 9 shows the longitudinal results for the 3CVT task. Cluster-based permutation analysis of within-subject changes revealed no significant differences between the baseline and follow-up visits in either the HC or MCI group. CBA measure for the second cluster significantly increased (p<0.05, ES = 0.25) in the MCI group and showed a non-significant decrease in the HC group (p = 0.052, ES = -0.26). CBA measure for the first cluster did not significantly change from baseline to follow-up visit. Fig 9B shows the test-retest reliability of cluster-based measures (ICC = 0.66, r = 0.51, p<0.001) and (ICC = 0.86, r = 0.76, p<0.001) for the first and second cluster, respectively. Performance measures did not significantly change from baseline to follow-up visit. The performance measures reliabilities were (ICC = 0.87, r = 0.76, p<0.001) and (ICC = 0.31, r = 0.28, p<0.05) for the reaction time and accuracy measure, respectively. Two participants changed their status from MCI (at their baseline visit) to healthy (at their follow-up visit). The baseline status was used for all the above analyses.

## EEG-based classification accuracy and repeatability

We trained and validated an SVM classifier to predict MCI diagnosis using the 12 predictors listed in Table 3. The re-substitution accuracy of the model trained and tested on the training data was 97% (AUC = 0.99). Fig 10A shows group difference between the MCI and HC groups in the training dataset with respect to the MCI classification scores (*ES = 3.1, df = 33, p<0.001*). Subsequent split-data cross-validation by testing this classifier on testing data set (at both baseline and follow-up visits) resulted in the classification performance (*AUC = 0.87, Accuracy 77%, balanced accuracy = 79%*) for the baseline data (Fig 10B) and (*AUC = 0.75, Accuracy 70%, balanced accuracy = 73%*) for the follow-up visit data (Fig 10C). The box plots

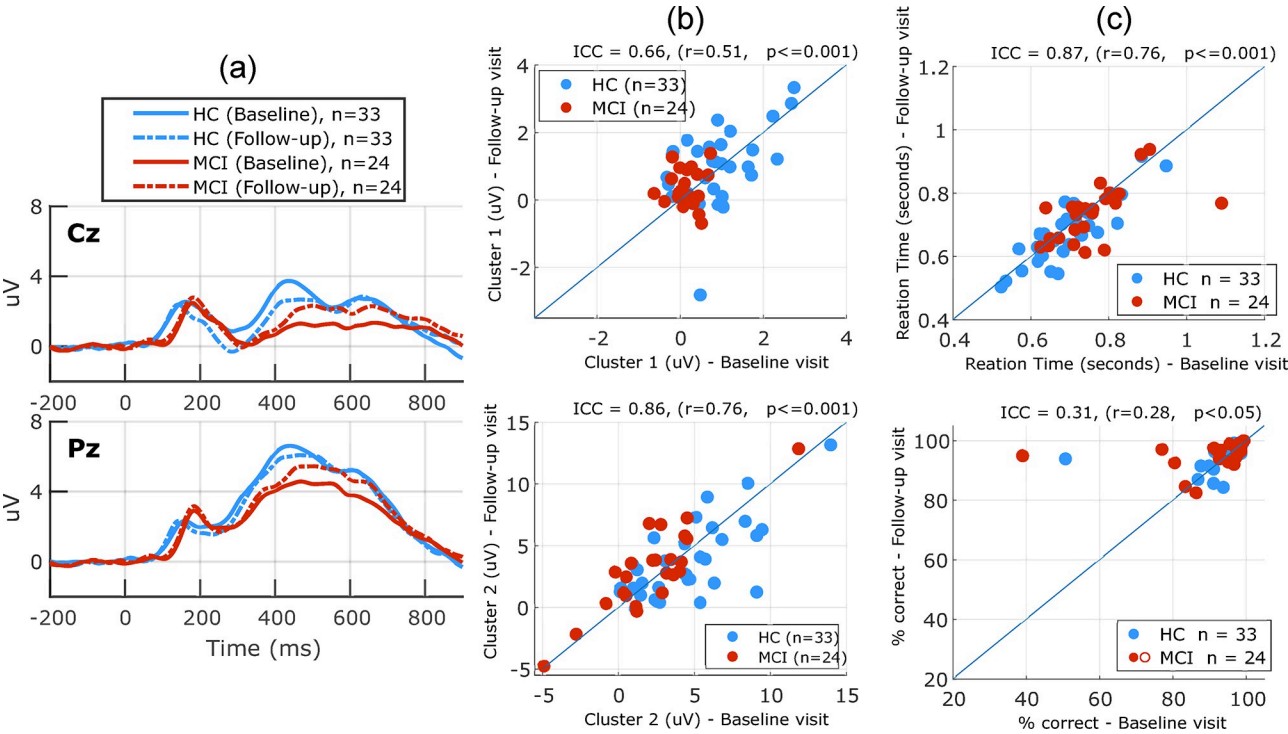

**Fig 9. Longitudinal changes in the 3CVT task.** (a) Grand average ERP waveforms (in response to frequent (Target) stimuli in 3CVT task) at both the baseline visit and 1-year follow-up, for the HC and MCI groups plotted for two EEG channels Cz, Pz as an example. (b) scatter plots showing the test-retest reliability of the EEG measures obtained using cluster 1 (top) and cluster 2 (bottom) in the 3CVT task. (c) scatter plots showing test-retest variability of the performance measures: Reaction time (top) and percent correct (bottom) in the 3CVT task.

in Fig 10 show the group differences between MCI and HC at the classifier operating point. Comparing the classification scores between baseline and follow-up visits in the testing data showed high consistency (*ICC = 0.84* (Cronbach's alpha), Pearson's correlation *r = 0.72*; *p<0.001*) (Fig 10D). The status of each participant at each visit (clinical diagnosis) was used for evaluating classifier performance on the corresponding data.

Furthermore, we also trained the classifier on all available baseline data (baseline data from either training or testing dataset) followed by a 5-fold cross validation. Classification performance after cross validation was similar (AUC = 0.89, Accuracy = 78%) (Fig 11B). The MCI classification score was significantly different between the MCI and HC group with a normalized effect size (ES = 1.95) (Fig 11A).

We assessed the predictive power of each individual predictor in the SVM classification model using two different methods. Table 4 shows the predictors ranked based on their importance (predictive power) using both the classifier weights (left) and the permutation feature methods (right) (refer to the methods section). The rankings were highly consistent between the two methods (Spearman's rho = 0.81) and show that the first 4 most predictive features are all EEG/ERP features (Resting state relative PSD cluster, 3CVT clusters 1/and 2, and SIR cluster1/or latency).

Finally, in order to explore the effect of the model type on the classification performance, we trained and tested a classifier using 5 different classification models (support vector machine, linear discriminant analysis, logistic regression, k-nearest neighbors, and decision tree). The performance of these classifiers after 5-fold cross validation are shown in the S1 Table. Unless otherwise reported, all models were trained and tested using default values of

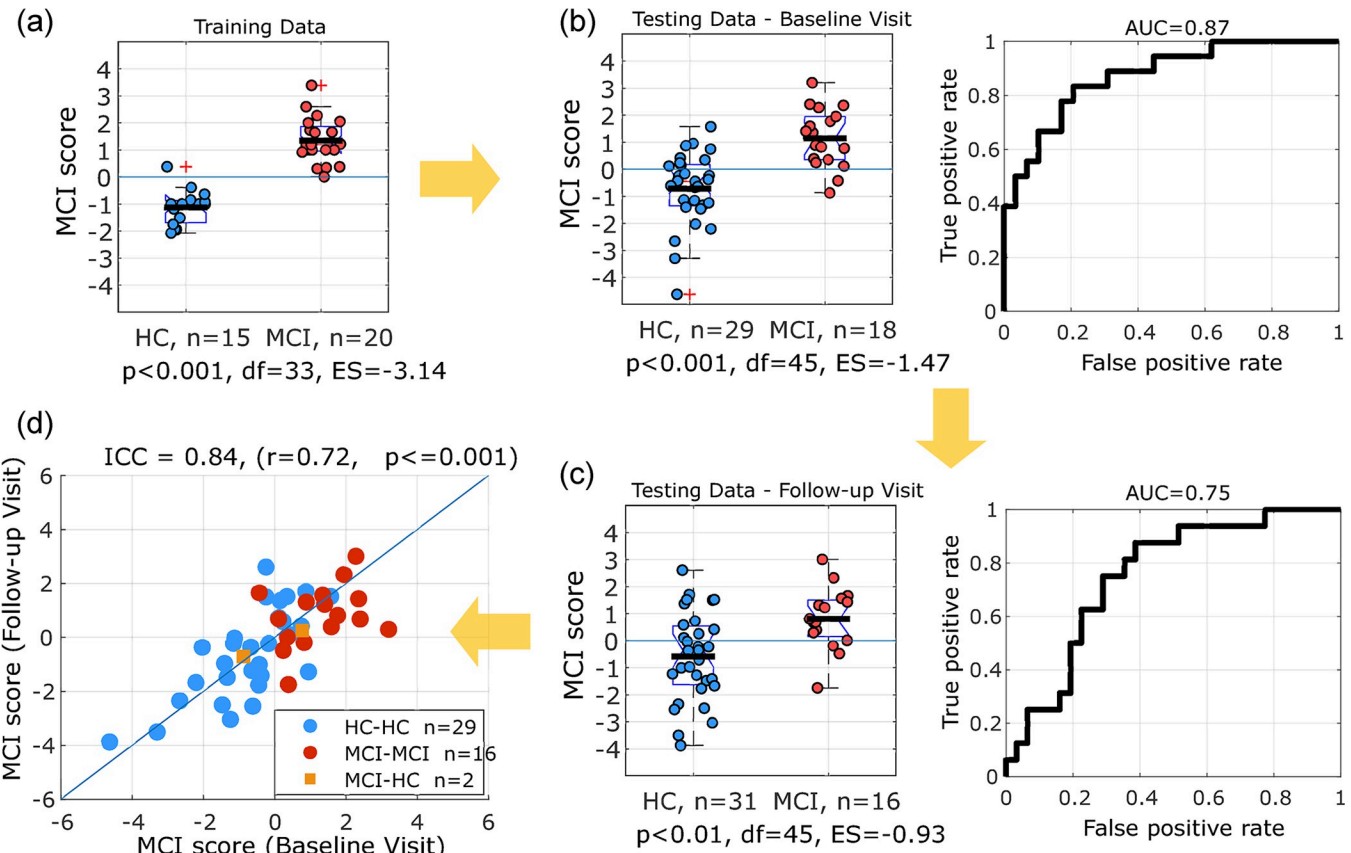

**Fig 10. Classification results.** (a) SVM classifier trained and overfitted on Training dataset, boxplots show the differences in the classification scores (MCI score) between the two groups (re-substitution classification accuracy = 97%, AUC = 0.99, balanced accuracy = 97%). (b) Performance of the same classifier on Testing dataset at baseline (AUC = 0.87, accuracy = 76%, balanced accuracy = 79%) and (c) Testing data at 1-year follow-up (AUC = 0.75, accuracy = 70%, balanced accuracy = 73%). (d) the scatter plot showing the correlation between MCI classification score at baseline and 1-year follow-up shows high consistency (ICC = 0.84, Pearson's r = 0.72). Two individuals were diagnosed as MCI at their baseline visit and later were deemed healthy at 1-year follow-up (denoted by MCI-HC).

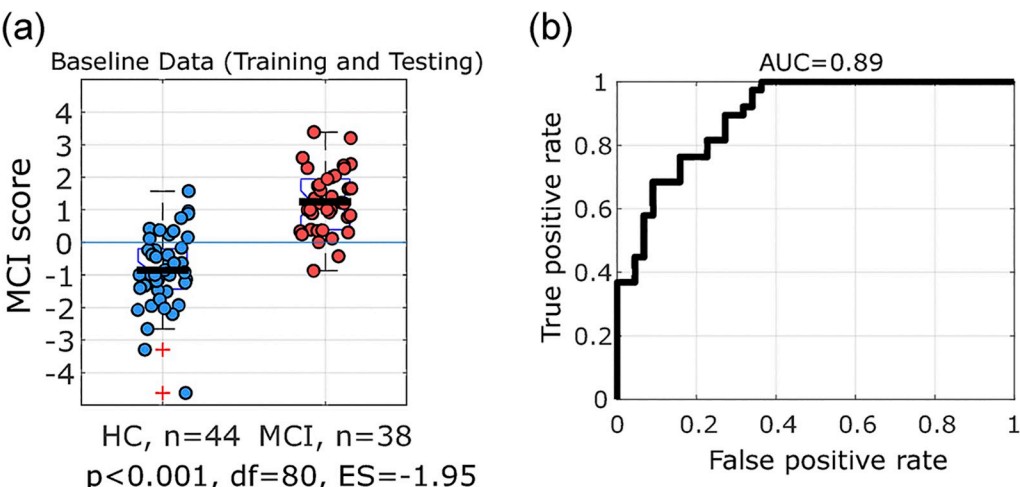

**Fig 11. Classification model using all data.** (a) Box plots of MCIs classification score using an SVM classifier trained on all available data at baseline visit. (b) Performance of this classifier after 5-fold cross validation (AUC = 0.89, accuracy = 77%).

**Table 4. Predictive power of individual predictors:** Predictors in the SVM classifier ranked based on their predictive power (from highest to lowest), using the relative weights in the SVM classifier (left), as well the % reduction in the performance based on permutation feature importance method (Right).

| Predictors ranked based on relative size of the weights in the SVM | | Predictors ranked based on permutation feature importance | | | |
|---|---|---|---|---|---|
| Predictor | SVM relative size of weights (%) | Predictor | % change (AUC) | % change (accuracy) | % change (balanced accuracy) |
| EEG-PSDrel | 16.72 | EEG-PSDrel | -4.24 | -6.67 | -6.88 |
| 3CVT-cluster-1 | 12.55 | 3CVT-cluster-2 | -3.08 | -5.83 | -5.85 |
| SIR-cluster-1 | 11.29 | 3CVT-cluster-1 | -3.04 | -4.76 | -4.93 |
| 3CVT-cluster-2 | 10.71 | SIR-Latency | -2.91 | -4.60 | -4.74 |
| SIR-Latency | 9.77 | * RT-SIR | -2.87 | -4.26 | -4.31 |
| * RT-SIR | 9.76 | SIR-cluster-1 | -2.29 | -3.13 | -3.38 |
| * RT-3CVT | 7.84 | * RT-3CVT | -1.5 | -2.33 | -2.44 |
| * PC-SIR | 6.97 | * PC-SIR | -0.86 | -2.20 | -2.34 |
| EEG-PSDabs | 6.96 | * PC-3CVT | -0.80 | -1.14 | -1.15 |
| 3CVT-Latency | 3.00 | EEG-PSDabs | -0.70 | -1.08 | -1.14 |
| SIR-cluster-2 | 2.49 | SIR-cluster-2 | -0.62 | -0.86 | -1.04 |
| * PC-3CVT | 1.93 | 3CVT-Latency | -0.40 | +0.03 | -0.15 |

* Behavioral performance-based predictors.

parameters in *Statistics and Machine Learning Toolbox* (version 11.4) in *Matlab® version 9.5 (R2018b)*.

We studied the prognostic value of the classification model by comparing the baseline value of the classification score against future cognitive decline as measured by MMSE change after 1-year. Fig 12 shows the scatterplots of baseline classification score against longitudinal

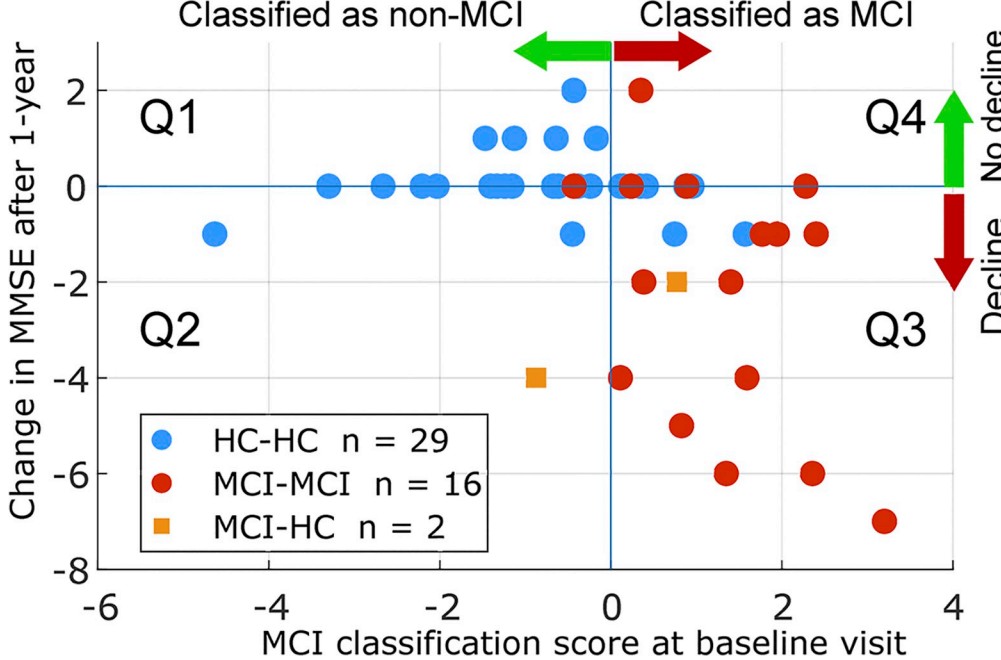

**Fig 12. Classification scores prognostic value.** Scatter plot of the changes in MMSE score of cognition after 1-year Follow-up (follow-up score–baseline score), versus the proposed MCI classification score at the baseline visit.

changes in MMSE. The space is divided into 4 quartiles (Q1: classified as healthy at baseline followed by no future decline in cognition, Q2: classified as healthy at baseline followed by decline in cognition after 1 year, Q3: classified as MCI at baseline followed by future decline in cognition, Q4: classified as MCI at baseline followed by no future decline in cognition). The percentage of participants in each quartile is 40%, 6%, 30%, and 23% in quartiles Q1 to Q4, respectively. Of those participants who were classified as MCI (Q3 and Q4), 56% showed signs of decline in cognition 1-year after while this percentage was 13% of participants who were classified as HC. Overall, decline in MMSE was correlated with the MCI classification score (r = 0.45, p<0.01).

## Discussion

We demonstrated the accuracy and reliability of a composite EEG/ERP biosignature of cognitive decline in individuals with MCI, derived from 12 individual electrophysiological biosignatures using machine learning classification. To our knowledge, this is the first study to present longitudinal assessment of the reliability of multi-modal EEG measures (including both resting state and ERP signatures) for quantifying cognitive decline. These EEG/ERP modalities included PSD of resting state EEG, ERP measures in two cognitive tasks focused on memory and attention, and the associated behavioral performance. We defined each standalone biosignature using a data-driven approach (cluster-based permutation analysis) to address multiplicity of comparisons in the channels/frequency/time space and to constrain the number of EEG/ERP biosignatures for each modality.

In resting state EEG (rsEEG), consistent with previous data [23,33,53–55] and compared to healthy controls, the MCI group showed elevated Theta (slower) and suppressed relative Beta (faster) EEG band frequencies, but no significant decline in Alpha power. In a previous work [23], we reported that individuals with AD exhibited a similar effect (albeit larger than that of MCI), as well as a decline in Alpha power that was not observed in MCI. We had also shown that in individuals with AD, Theta-to-Alpha ratio (TAR) is correlated with MMSE scores indicative of cognitive decline, suggesting TAR as a potential disease severity measure. However, in individuals with MCI, resting state EEG signatures incorporating Theta and Beta may be more sensitive than those incorporating Alpha power for tracking disease trajectory and conversion to dementia. Further prospective longitudinal studies are necessary to determine whether rsEEG measures alone could reliably track disease trajectory in MCI and AD dementia.

Here, in both ERP tasks and compared to HC, the MCI group exhibited delayed latency of the early component, reduced amplitude of the late component, and a decline in behavioral performance measured by accuracy and reaction time. These findings align with previous literature on ERP biomarkers of AD/MCI [41,56–59] and have been linked to an increased risk of MCI conversion to dementia [60]. Previous studies have utilized various ERP paradigms, including the classic oddball task, n-back task, Stroop task, and go/no go task. [59]. The cognitive ERP tasks in this study have previously demonstrated deficits in individuals with MCI [44], those under the acute effect of Cannabis [61], and individuals with HIV [45]. Our results are consistent with a recent systematic review of cognitive event-related potentials in MCI [59], indicating that decreased ERP amplitude and delayed latency are associated with AD-related pathology, though with less robust patterns in MCI [59]. In [59], the authors emphasized the importance of "reporting key contrast data or effect sizes" along with sensitivity and specificity. Biosignatures with substantial effect sizes and heightened sensitivity/specificity support the efficacy of these methods as biomarkers suitable for individual assessment alongside neuropsychological testing.

To that end, we compared individual EEG and ERP biosignatures by the normalized effect size of the difference between group means for the MCI and HC. ERP tasks-based biosignatures generally resulted in higher normalized effect sizes compared to resting state biosignatures, indicating better separability between the MCI and HC groups and higher efficiency as diagnostic biomarkers. Using 1-year follow-up data, we also assessed test-retest reliability of these biosignatures as candidate severity biomarkers for tracking the trajectory of cognitive deficits at an individual level. In individuals with MCI, cognitive deficits can manifest in various forms and cognitive domains. The ERP tasks in this study do not encompass all possible cognitive domains. However, we designed the SIR and 3CVT tasks specifically for assessing memory, attention, and executive function as these domains are reported to be affected earlier in the disease progression [40,62,63]. The individual EEG/ERP biosignatures exhibited various degrees of consistency across participants, suggesting they may reflect deficits in different cognitive domains.

In the present study we combined these individual biosignatures of MCI into a unified composite score (hereby denoted MCIsc). The composite score resulted in a higher normalized effect size (ES = 1.95) compared to standalone biosignatures. This method does not provide domain-specific measures of deficits but rather unifies these measures into an aggregated measure of cognitive decline. Large cohort studies with stratified patients based on impairment severity in specific domains may reveal direct association between individual predictors and specific cognitive domains.

Our MCIsc composite score was developed using a machine learning classifier. Hence, to demonstrate the validity of the MCIsc at an individual level, we split the data into disjoint training and testing sets to avoid model overfitting. Evaluating MCIsc at both visits on the testing data showed very high test-retest reliability (ICC = 0.84). Overall, these results support the first aim of this study and provide quantitative measures of robustness and efficiency for the multi-modal MCIsc composite score as a diagnostic biomarker of MCI cognitive deficits at an individual level.

Our findings and those of the prior studies demonstrate an associative rather than causative effect between cognitive decline and EEG/ERP signatures. As such, these significant differences do not necessarily rule out the possibility of indirect effects due to mediating and/or moderating variables. One such indirect effect could be mental fatigue [64], which has been reported to be higher in individuals with MCI and subjective cognitive decline [65,66]. Analysis of such effects is outside the scope of this paper and requires a different study design with assessment and analysis of fatigue as a covariate.

The second aim of this study was to assess the potential of MCIsc as a severity and prognostic biomarker for individuals with MCI. We hypothesized that greater severity of cognitive decline would imply a higher MCIsc and improved discrimination. Longitudinal testing and MMSE scores served as our disease severity measures. MCIsc showed a very high test-retest reliability. In grouped analyses of all participants (MCI and HC combined), MCIsc significantly correlated with MMSE scores. Additionally, changes in MCIsc significantly correlated with changes in MMSE, and MCIsc at baseline significantly correlated with subsequent changes in MMSE from baseline to follow-up. However, within the MCI group alone, similar but non-significant correlations were seen, possibly due to lack of MMSE diagnostic sensitivity for cognitive disorders [67]. Overall, while agreement between novel biomarkers of cognition and MMSE score is expected at both ends of the spectrum (severe cognitive decline versus completely healthy), using MMSE as ground truth for validating novel measures is not practically feasible due to the lack of adequate sensitivity and precision.

We evaluated the performance of the MCI classifier by comparing the classification scores against clinical diagnosis on a testing data set. However, MCI is both pathologically and

clinically heterogeneous, and a mismatch between the clinical diagnosis and the machine learning classification could be due to either modeling error or uncertainty and heterogeneity of MCI diagnosis. A false positive for an otherwise healthy individual could indicate subclinical neurophysiological deficits with increased risk factor for future decline while a false negative for an individual already diagnosed with MCI could indicate the lack of underlying pathology and a favorable prognosis. One of the limitations of the present work is that participants were diagnosed and stratified based on neuropsychological testing, without confirmed evidence of Alzheimer's pathology via diagnostic biomarkers. Not all individuals with MCI may have AD pathology, and only some progress to dementia [68,69]. For example, in a recent study of 331 participants with MCI [70], over half of the participants reverted to normal at 6 year follow-up, with some re-transitioning back to MCI at 12-year follow-up. In our study which had only 1 year between visits, 56% of those who were classified MCI by MCIsc exhibited decline in MMSE after 1-year, supporting the potential of MCIsc as a prognostic biomarker.

In the present work, MCIsc is based on an operating point of the classifier that maximized accuracy in predicting MCI status and has higher sensitivity than specificity. In various clinical and research settings and applications, different tradeoffs between sensitivity and specificity may be desirable. For example, in clinical trials recruiting patients at early stages of MCI due to AD, MCIsc could be used as a cost-effective screening tool, at an operating point that provides higher specificity, or otherwise an optimized tradeoff between sensitivity and specificity. In clinical practice, a combination of MCIsc scores with clinical factors could inform physicians and their patients regarding the need for further clinical evaluation.

The MCIsc in this work resulted from a model, trained on data that was limited to MCI and healthy controls. Future work is needed to evaluate the specificity of this measure to etiology of cognitive decline. For example, given the similarities and differences between biosignatures of cognitive decline in various neurodegenerative conditions (e.g. Alzheimer versus Lewy body dementia [71]), a similar machine learning approach could be applied to differentiate between MCI associated with various pathologies.

Moreover, it is important to differentiate between biological and clinical staging of neurodegenerative diseases. In the United States, recent research community consensus favors a biological definition of Alzheimer's disease regardless of clinical symptoms [36]. Some individuals with demonstrated pathology may remain asymptomatic, possibly due to cognitive or other resilience factors remaining to be characterized, while others may decline quickly. Our proposed method aimed at quantifying cognitive symptoms using neural underpinnings of cognitive function. However, it is not clear how compensatory brain mechanisms in individuals with AD pathology may impact MCIsc. Nonetheless, this approach and method can complement biological staging and potentially help characterize disease trajectory.

Our approach may also have future applications in clinical management and monitoring during administration of disease modifying therapies. For example, anti-amyloid monoclonal antibodies (mAbs) [14,72,73] can reduce or clear amyloid pathology and possibly slow or partially reverse pathological and clinical progression. However, these therapies have only thus far demonstrated a modest effects on cognitive functioning and have potential safety issues [14], suggesting an urgent need for expeditious and scalable methods that can inform clinicians on treatment decisions at an individual level. Studies have reported a "non-homogenous" treatment effect and covariates-treatment interactions [74,75] that may warrant a personalized medicine approach [76] based on individual treatment response (ITR). Unlike cognitive rating scales currently in use (e.g. Clinical Dementia Rating Scale [12], MoCA, MMSE, etc.), our proposed method can potentially provide a more sensitive measure of treatment effectiveness at an individual level.

In summary, we introduced a quantitative EEG/ERP biosignature for monitoring cognitive decline in Mild Cognitive Impairment (MCI). Our method shows promise as a cost-effective, non-invasive, scalable diagnostic tool of MCI at an individual level, paving the road for a precision medicine approach in neurodegenerative diseases. Future research is needed to confirm the method's potential for differential diagnosis and long-term prognosis in MCI.

## Supporting information

**S1 Fig. SIR task grand average ERP waveforms at all EEG channels, baseline visit, plotted for novel stimuli in HC and MCI groups.**
(TIF)

**S2 Fig. 3CVT task grand average ERP waveforms at all EEG channels, baseline visit, plotted for target stimuli in HC and MCI groups.**
(TIF)

**S1 Table. Other classification models: Comparing classification performance for different types of classification models trained on baseline dataset (n = 82) and after 5-fold cross validation.**
(DOCX)

## Author Contributions

**Conceptualization:** Amir H. Meghdadi, Ajay Verma, Chris Berka.

**Data curation:** Amir H. Meghdadi.

**Formal analysis:** Amir H. Meghdadi.

**Funding acquisition:** David Salat, Joanne Hamilton, Bradley F. Boeve, Erik K. St Louis, Chris Berka.

**Investigation:** Amir H. Meghdadi, David Salat, Joanne Hamilton, Yue Hong, Bradley F. Boeve, Erik K. St Louis, Chris Berka.

**Methodology:** Amir H. Meghdadi, Yue Hong.

**Project administration:** Chris Berka.

**Resources:** David Salat, Joanne Hamilton, Bradley F. Boeve, Erik K. St Louis, Chris Berka.

**Software:** Amir H. Meghdadi.

**Supervision:** Amir H. Meghdadi, Chris Berka.

**Validation:** Amir H. Meghdadi.

**Visualization:** Amir H. Meghdadi.

**Writing – original draft:** Amir H. Meghdadi.

**Writing – review & editing:** Amir H. Meghdadi, Erik K. St Louis, Chris Berka.

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
