## [Decision Letter · Decision Letter 0]

28 May 2024

PONE-D-24-00944EEG and ERP biosignatures of mild cognitive impairment for longitudinal monitoring of early cognitive decline in Alzheimer’s diseasePLOS ONE

Dear Dr. Meghdadi,

Thank you for submitting your manuscript to PLOS ONE. After careful consideration, we feel that it has merit but does not fully meet PLOS ONE’s publication criteria as it currently stands. Therefore, we invite you to submit a revised version of the manuscript that addresses the points raised during the review process.

We look forward to receiving your revised manuscript.

Kind regards,

Stephen D. Ginsberg

Section Editor

PLOS ONE

Journal Requirements:

[This work is supported by National Institute of Health Grant numbers R44AG050326, R44AG054256 as well as NIA ADRC grant numbers P30 AG062677.]

 [This work is supported by National Institute of Health Grant numbers R44AG050326, R44AG054256 as well as NIA ADRC grant numbers P30 AG06267]

 [This work is supported by National Institute of Health Grant numbers R44AG050326, R44AG054256 as well as NIA ADRC grant numbers P30 AG062677].  

5. Thank you for uploading your study's underlying data set. Unfortunately, the repository you have noted in your Data Availability statement does not qualify as an acceptable data repository according to PLOS's standards.

6. Please amend the manuscript submission data (via Edit Submission) to include author Doris Hong.

8. We notice that your supplementary figures are uploaded with the file type 'Figure'. Please amend the file type to 'Supporting Information'. Please ensure that each Supporting Information file has a legend listed in the manuscript after the references list.

9. "We notice that your supplementary tables are included in the manuscript file. Please remove them and upload them with the file type 'Supporting Information'. Please ensure that each Supporting Information file has a legend listed in the manuscript after the references list.

10. Please include captions for your Supporting Information files at the end of your manuscript, and update any in-text citations to match accordingly. Please see our Supporting Information guidelines for more information: http://journals.plos.org/plosone/s/supporting-information.

Reviewers' comments:

Reviewer's Responses to Questions

**Comments to the Author**

1. Is the manuscript technically sound, and do the data support the conclusions?

Reviewer #1: Yes

Reviewer #2: Partly

2. Has the statistical analysis been performed appropriately and rigorously? 

Reviewer #1: Yes

Reviewer #2: Yes

3. Have the authors made all data underlying the findings in their manuscript fully available?

Reviewer #1: Yes

Reviewer #2: Yes

4. Is the manuscript presented in an intelligible fashion and written in standard English?

Reviewer #1: Yes

Reviewer #2: Yes

5. Review Comments to the Author

Reviewer #1: The intriguing study used resting-state EEG and cognitive ERP data in 46 normal older adults and 43 age-matched individuals with mild cognitive impairments (MCIs) to predict cognitive decline risk. Notably, task-related EEG exhibited a higher normalized effect size compared to resting-state EEG signatures. The study sets itself apart from many similar investigations with several major strengths, including the creation of composite biomarkers from resting EEG and tasks evoked potentials, utilization of one-year follow-up data to assess test-retest reliability of the biosignatures for cognitive decline at the individual level, and the incorporation of a composite EEG/ERP biosignature of cognitive decline with various mental scores to evaluate the severity of individuals with MCI. Overall, the manuscript should contribute to the literature by providing much-needed insights into the development of low-cost cognitive screening tools.

A suggestion that better justification should be provided for the selection of tasks for the model, cognitive motor performance, and corresponding EEG features. It is essential to recognize that MCI cognitive impairments often manifest in different forms, such as deficits in memory, language, executive function, or visual-spatial abilities.

A limitation of the model that warrants acknowledgment in the discussion is that the neurocognitive scores included in the model are limited to MMSE, which may not be sufficiently discriminative.

A minor issue arises in Table 2, where the demographic table for participants who had complete data at the follow-up visit is mentioned multiple times before it is ultimately presented.

Reviewer #2: The authors developed a unified measure of cognitive decline using multiple EEG modalities, including resting-state EEG and event-related potential (ERP) tasks. Comparing individuals with mild cognitive impairment (MCI) with healthy controls, the MCI group showed distinct EEG patterns and reduced performance on ERP tasks. A support vector machine classifier achieved high accuracy in predicting MCI. The classification scores were consistent over time, indicating reliable test-retest results. This demonstrates the potential of EEG/ERP for early diagnosis and monitoring of Alzheimer's disease in clinical trials. However, the manuscript needs to be improved.

Please clarify the following points

0) Patients with cognitive decline are likely to be fatigued when doing the three tasks (resting, 3CVT and SIR). Please clarify whether the study conducted a randomisation of tasks in order to avoid possible fatigue effects on the SIR which being the last one is the most prone to such effects. If the task sequence was not randomized then all results could be influenced by fatigue. In other words, how much confidence do you have that the SIR results are specific and not confounded by the protocol?

1) How many trials were used to calculate ERP in both 3CVT and SIR, and with regard to which type of stimulus and type of response (correct/wrong)? And after rejection of the trials, how many patients/hs were adopted?

2) Classification

The classification of MCIs needs to be made more robust. First of all, a comparison of accuracy values against a dummy classifier should be indicated or a balanced accuracy should be used. I

Secondly, it should be estimated whether the most predictive variables are neural or behavioural variables.

Furthermore, neural variables associated with tasks are latencies, when the first-order effect of potentials is absorbed in voltage attenuation. Comment on why features were not extracted directly from amplitude instead of looking at latency. The cluster did something similar related to amplitude, but it is not clear which amplitude we are dealing with in Figure 4D/5D. The neural features associated with the rest instead are the PSDs. Why was it not decided to study PSDs in ERPs as well?

Furthermore, what role do mismatches play? For example, an HS classified as MCI could have latent development of a pathology, whereas MCI classified as HS could have a compensatory dynamic in place. So how much of the mismatch can be attributed to errors in the model or to actual prediction?

3) Comparison with the literature

It seems to me that this work is very much in line with the work of Shani Waninger 2018 ( https://doi.org/10.1016/j.dadm.2018.05.007).

Comment on what you add and where you replicate and where you differ from that work.

4) outliers

Authors say they did not exclude behavioral outlier data to replicate variability in the real world. Please comment on why they support this position and whether they also did the analyses excluding behavioral outliers (eg excluding >3 sigma values in RT or accuracy) and in that case whether these are very different from those with outliers included.

6. PLOS authors have the option to publish the peer review history of their article (what does this mean?). If published, this will include your full peer review and any attached files.

Reviewer #1: No

Reviewer #2: No

---

## [Author Response · Author response to Decision Letter 0]

7 Jul 2024

We would like to thank the reviewers for their thoughtful comments and suggestions. We have revised the manuscript accordingly. We have described the revisions and answered the reviewers comments in the Response to Reviewers comments file uploaded via the submission system. Here is a copy-paste of the response for convenience. 

Reviewer #1: The intriguing study used resting-state EEG and cognitive ERP data in 46 normal older adults and 43 age-matched individuals with mild cognitive impairments (MCIs) to predict cognitive decline risk. Notably, task-related EEG exhibited a higher normalized effect size compared to resting-state EEG signatures. The study sets itself apart from many similar investigations with several major strengths, including the creation of composite biomarkers from resting EEG and tasks evoked potentials, utilization of one-year follow-up data to assess test-retest reliability of the biosignatures for cognitive decline at the individual level, and the incorporation of a composite EEG/ERP biosignature of cognitive decline with various mental scores to evaluate the severity of individuals with MCI. Overall, the manuscript should contribute to the literature by providing much-needed insights into the development of low-cost cognitive screening tools.

A suggestion that better justification should be provided for the selection of tasks for the model, cognitive motor performance, and corresponding EEG features. It is essential to recognize that MCI cognitive impairments often manifest in different forms, such as deficits in memory, language, executive function, or visual-spatial abilities.

Thank you for raising the issue of heterogeneity of MCI in terms of the affected cognitive domains. In the revised version of the manuscript, we acknowledge this issue in the discussion section. To answer the specific question, we acknowledge that our tasks are not exhaustive in terms of covering all these domains. However, we explained the rational for choosing the SIR and 3CVT task in this study. Analysis of the correlation between individual EEG/ERP predictors (Figure 6) shows correlation between some and independence between other individual predictors supporting the idea that different EEG/ERP biosignatures may be associated with different aspects of cognition. Future work may include subgrouping MCI patients based on the affected domain and studying the association between each individual EEG signature and the cognitive domain. For now, this remains one of the limitations of this study. 

A limitation of the model that warrants acknowledgment in the discussion is that the neurocognitive scores included in the model are limited to MMSE, which may not be sufficiently discriminative.

Yes, we agree with the reviewer on the inadequacy and lack of sensitivity of MMSE and we have acknowledged this limitation of MMSE with citing at least one reference as well (please see the discussion section and reference [67] in the revised manuscript. However, note that we have not used MMSE for training our model. Rather, we have compared changes in MMSE against our model predictions for better validation. We have elaborated this limitation further in the revised manuscript. Please refer to the discussion section. 

A minor issue arises in Table 2, where the demographic table for participants who had complete data at the follow-up visit is mentioned multiple times before it is ultimately presented.

Thank you. We fixed this by moving Table 2 to immediately after the paragraph that first refers to it. 

Reviewer #2: The authors developed a unified measure of cognitive decline using multiple EEG modalities, including resting-state EEG and event-related potential (ERP) tasks. Comparing individuals with mild cognitive impairment (MCI) with healthy controls, the MCI group showed distinct EEG patterns and reduced performance on ERP tasks. A support vector machine classifier achieved high accuracy in predicting MCI. The classification scores were consistent over time, indicating reliable test-retest results. This demonstrates the potential of EEG/ERP for early diagnosis and monitoring of Alzheimer's disease in clinical trials. However, the manuscript needs to be improved.

Please clarify the following points

0) Patients with cognitive decline are likely to be fatigued when doing the three tasks (resting, 3CVT and SIR). Please clarify whether the study conducted a randomisation of tasks in order to avoid possible fatigue effects on the SIR which being the last one is the most prone to such effects. If the task sequence was not randomized then all results could be influenced by fatigue. In other words, how much confidence do you have that the SIR results are specific and not confounded by the protocol?

Thank you for raising this issue. The task sequence was NOT randomized. We agree with the reviewers that fatigue could potentially affect the task performance and endpoints. However, the study design has been based on a fixed order of tasks to keep any potential effect the same across all study participants. While randomized (AND counter-balanced) order of tasks was possible, we would have likely needed more data to perform the analysis with the same statistical power, given the extra confounding factor (order of task).

We believe the effect of fatigue has likely contributed to the fact that only 60% of MCI participants completed all the tasks (while this percentage in the HC group was 97%, refer to the data quality section. Nonetheless, participants who did complete the SIR task showed significant ERP deficits as reported in the paper. 

Inspired by the reviewer’s comment on this issue, we explored this topic further by analyzing the correlation between ERP predictors in the SIR task and behavioral performance. We assumed that behavioral performance in the SIR task (reaction time and accuracy of response) are likely affected by fatigue, so if the ERP deficits (measured by cluster 1 and cluster 2 values) are merely the effect of more fatigue in the MCI group and not pathological signatures, we would expect significant correlation between these ERP predictors and behavioral performance. To that end we computed the correlation coefficient (/plotted scatter plots) for correlation between these ERP predictors and behavioral performance for each of the MCI and HC groups separately (See supporting document: response_to_reviewers_fatigue_in_SIR.pdf). 

Overall, there was no statistically significant correlation between the measures. Closer examination of the findings shows that while in the HC group there was a non-significant (p=0.13, p=0.11) trend for negative correlation (r=-0.23, r=-0.24) between reaction time and cluster1, cluster 2 predictors, respectively, there was no such correlation for the MCI group (r=-0.05, r=-0.09). In short, ERP deficits in the MCI group were not associated with performance measures and thus are unlikely to significantly be associated with the level of fatigue. Similar analysis for accuracy measure (percent of trials with correct response), showed no significant correlation. The most notable trend observed for association between percent correct response (PC) and cluster 2 deficits in the MCI group (r=0.26, p=0.097). However, that trend also existed in the HC group (r=0.2, p=0.188). 

Overall, we believe that while individuals with MCI are more likely to be fatigued after a long session and while fatigue could affect EEG/ERP signatures, the reported neurophysiological deficits in MCI are less likely to be neurophysiological effects of the elevated fatigue and more likely to be neurophysiological effects of the underlying neural disfunctions. Nonetheless, it is advisable that these tests be conducted under similar circumstances as much as possible to minimize the effect of confounders such as fatigue (e.g. patients may be recommended to perform the task in the rested and fed condition). In the present study patients were scheduled in the mornings. 

1) How many trials were used to calculate ERP in both 3CVT and SIR, and with regard to which type of stimulus and type of response (correct/wrong)? And after rejection of the trials, how many patients/hs were adopted?

All ERP results were computed/reported on trials that were responded correctly. After excluding the incorrect response and rejected-for-data-quality, the number of ERP trials that were used for subsequent analyses for each task were as follows: 

median, mean, range (min-maxis)

SIR novel stimuli: 62, 56.3, (9-78)

3CVT target stimuli: 215, 198.2, (14-249)

The number of participants adopted for analysis are reflected in Table 1 and 2 and Figures 4 and 5. Overall, in the SIR task and at the baseline visit n=46 and n=43 participants were adopted for analysis after all exclusions (in the HC and MCI groups, respectively). In the 3CVT task, n=50 and n=48 participants were adopted for analysis. 

2) Classification

The classification of MCIs needs to be made more robust. First of all, a comparison of accuracy values against a dummy classifier should be indicated or a balanced accuracy should be used. I

We have updated the manuscript and reported both accuracy and balanced accuracy of the classification results. The differences were not remarkable. 

Secondly, it should be estimated whether the most predictive variables are neural or behavioural variables.

Inspired by this interesting comment, we investigated the predictive power of all the features and updated both the methods and the results section (Table 4). We used two different methods both based on the classifier weights and by a feature permutation method (refer to the methods section) and ranked the 12 predictors based on their predictive powers. In both cases, the behavioural variables were not among the most predictive variables. Furthermore, our experimental analyses have shown that while behavioural variables alone might be useful in yielding acceptable classification accuracies, such classifiers would be less reliable (test-retest reliability) and in principle also less specific (they can be more affected by other confounding factors). 

Furthermore, neural variables associated with tasks are latencies, when the first-order effect of potentials is absorbed in voltage attenuation. Comment on why features were not extracted directly from amplitude instead of looking at latency. The cluster did something similar related to amplitude, but it is not clear which amplitude we are dealing with in Figure 4D/5D. The neural features associated with the rest instead are the PSDs. Why was it not decided to study PSDs in ERPs as well?

Thank you for this comments. Please note that in both tasks (3CVT and SIR), BOTH amplitudes and latencies have been used as predictors. Looking at the 12 predictors, both cluster 1 and cluster 2 are defined based on the amplitude differences between the two groups while SIR-Latency and 3CVT-Latency are merely based on the latency differences. Arguably cluster 1 differences (in both 3CVT and SIR) are just another manifestation of the latency shift (judging by the overlap between the time window and the direction of the effect), while cluster 2 difference are mainly derived by amplitude reduction. In summary examining ERP waveforms reveals both a latency shift and an amplitude reduction in the ERP waveforms. We have quantified the latency shifts both directly (by measuring the peaks) and indirectly emerging from cluster-based analysis. We have also quantified amplitude reductions by the emerging cluster 2 in both tasks. 

Furthermore, what role do mismatches play? For example, an HS classified as MCI could have latent development of a pathology, whereas MCI classified as HS could have a compensatory dynamic in place. So how much of the mismatch can be attributed to errors in the model or to actual prediction?

Thanks for raising this important topic. As the reviewer correctly pointed out, a discrepancy between a clinician judged diagnosis and the machine learning classification score could be due to two different reasons. One is the modeling error (that can be only avoided by more data and higher data quality), and another is the underlying pathological reasons that can cause

 At this point and given the heterogeneity of MCI as a group, it is difficult to distinguish imperfection of the model (modeling error) from. Other than longitudinal analysis, 

Regardless of that distinction, currently at the optimal operating point of the classifier, it has higher sensitivity than specificity resulting in more false-positive MCI than false-negatives. This might be a preferred bias depending on the applications. For patient screening at the health care level, positive scores could trigger decisions for further evaluation and close monitoring of the patients especially if it is accompanied by patients’ subjective cognitive complaints. On the other hand, for patient screening in clinical trials, where the goal is to recruit patients with more likelihood of being on the path to AD, it may be more preferred to adjust the classifier operating point for achieving higher specificity to be more aligned with the inclusion criteria. 

3) Comparison with the literature

It seems to me that this work is very much in line with the work of Shani Waninger 2018 ( https://doi.org/10.1016/j.dadm.2018.05.007).

Comment on what you add and where you replicate and where you differ from that work.

Yes, Waninger 2018 is one of our earliest works on this topic. There are many difference between the two papers that make them substantially difference other than the fact that that was a different study on a different cohort. 

A) the 2018 paper is only a cross-sectional study whereas the present paper has both cross sectional and longitudinal component 

B) The 2018 paper has only two ERP tasks (3CVT and SIR) whereas the current has findings in resting state as well. Moreover, the 3CVT and SIR in the current study have slightly difference ISI parameters (slower) to accommodate individuals with cognitive decline.

C) The 2018 paper presents only univariate parametric testing whereas the current paper provides a comprehensive cluster-based permutation analysis

D) The 2018 paper provides only group level differences and no machine learning, whereas the current paper is focused on findings at an individual level with machine learning models for assessment and prediction. 

Overall, we have replicated the reduction in LPP component in both tasks. The latency differences in the 2018 study were unremarkable and not reported. 

4) outliers

Authors say they did not exclude behavioral outlier data to replicate variability in the real world. Please comment on why they support this position and whether they also did the analyses excluding behavioral outliers (eg excluding >3 sigma values in RT or accuracy) and in that case whether these are very different from those with outliers included.

We agree that removing outliers is a delicate topic and should be considered carefully. True outliers (data entry error, technical issues, and measurement errors) are always expected to be removed of course. However, unusually high or low values might be due to various reasons. Since this paper is focused on the robustness of the findings at an “individual level”, we decided to keep the outliers in, as it demonstrates how often unusual values might be observed and how they affect the assessments. Nonetheless and inspired by the reviewer’s comment we reviewed all the data both by visual inspection and by computing z-core values of the outliers and investigated the effect of removing outliers. The predictors that were deemed as outliers (zsc>4 and visually confirmed as outlier) affected 2 HC and 2 MCI participants as follows:

Cluter-1 EEG data for one HC subject at follow-up visit only (Figure 9b)

Accuracy of the 3CVT test (behavioral) for one HC and two MCI individuals at baseline (Figure 5f and Figure 9c). 

For the group analysis exclusion of the outliers at baseline (extremely low accuracy in the 3CVT task) did not significantly change the effect size or the p values. We re-analyzed the classification of the participants excluding the outliers. Similarly, the results in Figure 10d did not change in terms of correlation between base

---

## [Editor Report · Decision Letter 1]

15 Jul 2024

EEG and ERP biosignatures of mild cognitive impairment for longitudinal monitoring of early cognitive decline in Alzheimer’s disease

PONE-D-24-00944R1

Dear Dr. Meghdadi,

We’re pleased to inform you that your manuscript has been judged scientifically suitable for publication and will be formally accepted for publication once it meets all outstanding technical requirements.

Kind regards,

Stephen D. Ginsberg, Ph.D.

Section Editor

PLOS ONE

---

## [Editor Report · Acceptance letter]

30 Jul 2024

PONE-D-24-00944R1 

PLOS ONE

Dear Dr. Meghdadi, 

I'm pleased to inform you that your manuscript has been deemed suitable for publication in PLOS ONE. Congratulations! Your manuscript is now being handed over to our production team.

Kind regards, 

on behalf of

Dr. Stephen D. Ginsberg 

Section Editor

PLOS ONE